# GroEL/ES chaperonin unfolds then encapsulates a nascent protein on the ribosome

Alžběta Roeselová [1], Sarah L. Maslen[2], Jessica Zhiyun He[3], Gabija Jurkeviciute[3], Aleksandra Pajak[1], J. Mark Skehel[2], Radoslav I. Enchev [3] & David Balchin [1]✉

The bacterial chaperonin GroEL/ES promotes protein folding post-translation by transiently encapsulating client proteins within a central chamber. GroEL also binds translating ribosomes in vivo, implying an additional role in cotranslational folding. However, how GroEL/ES recognises and modulates ribosome-tethered nascent proteins is unclear. Here, we used biochemical reconstitution, structural proteomics and electron microscopy to study the mechanism by which GroEL/ES engages nascent polypeptides. We show that GroEL binds nascent chains on the inside of its cavity via the apical domains and disordered C-terminal tails, resulting in local structural destabilization of the client. Ribosome-tethered nascent domains are partially encapsulated upon GroES binding to GroEL, and recover their original conformation in the chaperonin cavity. Reconstitution of chaperone competition at the ribosome shows that both Trigger factor and GroEL can be accommodated on long nascent chains, but GroEL and DnaK are mutually antagonistic. Our findings extend the role of GroEL/ES in de novo protein folding, and reveal a plasticity of the chaperonin mechanism that allows cotranslational client encapsulation.

The chaperonins are essential components of the cellular chaperone network in all domains of life[1]. They form large cage-like structures in which client proteins are temporarily encapsulated[2]. The best characterised chaperonin is *E. coli* GroEL/ES. GroEL subunits assemble into two heptameric rings arranged back-to-back, which are transiently capped by heptameric GroES in an ATP-driven conformational cycle (Fig. 1a)[3,4]. GroEL binds folding intermediates on the inside of the open ring. ATP binding to GroEL then induces a conformational change that allows binding of GroES, which displaces the client into the newly-formed GroEL/ES cavity[5–8]. GroEL and GroES monomers are expressed at approximately stoichiometric levels[9], and both asymmetric (one GroES oligomer per GroEL double-ring) and symmetric (two GroES per GroEL) complexes coexist in vivo[10]. Thus, one or both cavities can be folding-active. Encapsulation isolates clients from inter-molecular interactions, and in some cases actively promotes substrate folding[2].

Obligate clients often critically require the chaperonin to reach their native conformation on a physiologically-relevant timescale[11–14].

GroEL/ES binds newly-synthesised proteins in vivo[15,16], and is part of a larger chaperone network that dynamically partitions between the ribosome and bulk cytosol. The most abundant components are Trigger factor (TF), which has direct affinity for ribosomes, and DnaK (Hsp70)[1]. GroEL overexpression can partially compensate for the loss of TF and DnaK in *E. coli*[17,18], suggesting partial functional redundancy. Although GroEL/ES most prominently functions post-translation and downstream of TF and DnaK[19], it also acts directly at the ribosome. GroEL binds translating ribosomes in vitro[20,21] and is recruited to a subset of nascent chains (NCs) in vivo[22,23]. At least 50 different NCs bind GroEL, representing ~10% of all GroEL clients[22–25]. GroEL has also been shown to promote septin synthesis in bacteria[26], and alter the folding of nascent DHFR[27].

[1]Protein Biogenesis Laboratory, The Francis Crick Institute, London, UK. [2]Proteomics Science Technology Platform, The Francis Crick Institute, London, UK. [3]Visual Biochemistry Laboratory, The Francis Crick Institute, London, UK. ✉e-mail: david.balchin@crick.ac.uk

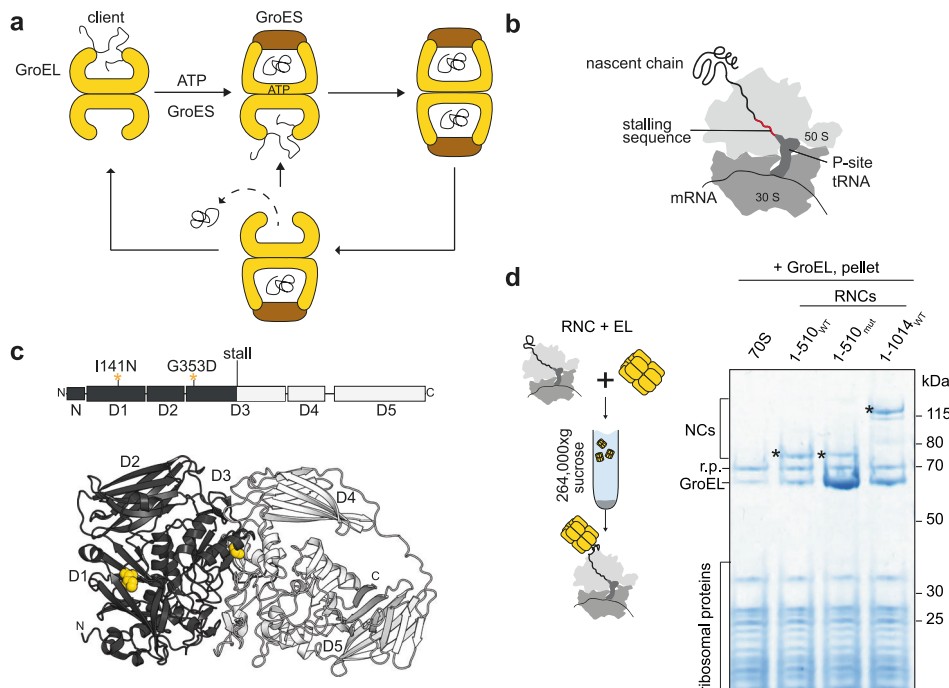

**Fig. 1 | GroEL binds ribosome:nascent chain complexes. a** Simplified functional cycle of GroEL/ES. **b** Stalled 70S ribosome:nascent chain complex (RNC). **c** Domain organisation and monomer structure (PDB: 6CVM) of *E. coli* β-galactosidase (β-gal). Positions of destabilising mutations I141N and G353D are shown in yellow. The first 510 residues of β-gal, corresponding to RNC$_{1-510}$, are coloured black. **d** GroEL binds preferentially to a conformationally destabilised RNC. Left: schematic of co-sedimentation assay. Right: Coomassie-stained SDS-PAGE of the resuspended ribosomal pellet. Prior to sedimentation, GroEL was incubated with either empty ribosomes (70S), wild-type RNC$_{1-510}$ (1-510WT), mutated (I141N, G353D) RNC$_{1-510}$ (1-510$_{mut}$), or RNC$_{1-1014}$ (1-1014$_{WT}$). RNCs were purified from Δ*tig* cells. Bands corresponding to the NCs (*), GroEL and ribosomal proteins (r.p.) are indicated. The experiment was repeated three times with similar results. Source data are provided as a Source Data file.

The post-translational function of GroEL/ES has been extensively characterised using chemically-denatured, full-length client proteins. Unlike these models, partially-synthesised NCs lack complete sequence information, are sterically influenced by proximity to the ribosome surface, and can populate unique structures compared to bulk solution[28–33]. How GroEL binds and modulates cotranslational folding intermediates is unclear. Moreover, ribosome-tethered NCs substantially exceed the estimated ~70 kDa size-limit of the chaperonin cavity[8], raising the question of whether cotranslational clients can be encapsulated by GroEL/ES.

Here, we address the mechanism of chaperonin function during cotranslational folding. We show that ribosome-tethered polypeptides bind the inside of the GroEL cavity and are partially encapsulated by GroEL/ES. The NC is locally destabilised by GroEL binding, but recovers its original conformation upon encapsulation. Our data suggest a role for GroEL/ES in rescuing cotranslational misfolding, and indicate that clients can benefit from confinement in the chaperonin cavity before they are released from the ribosome.

## Results

### Reconstitution of GroEL complexes with ribosome-bound nascent polypeptides

We first sought to establish a system for studying GroEL binding to nascent polypeptides. To generate homogeneous cotranslational chaperone clients, we purified stalled ribosome:nascent chain complexes (RNCs) from *E. coli*, each consisting of a defined NC sequence stably tethered to the ribosome (Fig. 1b). As an NC model we chose the large multidomain protein *E. coli* β-galactosidase (β-gal) (Fig. 1c). β-gal is expected to populate structurally diverse cotranslational folding intermediates during vectorial synthesis, affording the opportunity to sample different binding scenarios. Moreover, we previously found that β-gal RNCs frequently co-purified with sub-stoichiometric amounts of

GroEL[31]. Here, to study binding under controlled conditions, we reconstituted GroEL:RNC complexes using purified components in vitro. To facilitate accurate quantification of GroEL/GroES, we used tryptophan mutants which retained wild-type chaperone function (Supplementary Fig. 1a, b). We found that GroEL co-sedimented with several β-gal RNCs, indicative of stable complex formation (Fig. 1d and Supplementary Fig. 1c). GroEL also co-sedimented with empty 70S ribosomes, albeit to a lesser extent, suggesting a weak interaction with ribosomes even in the absence of NC. We found that GroEL bound to RNC$_{1-510}$ which exposes 2½ domains of β-gal. Binding was further stabilised by introducing NC mutations that disrupt the fold of domains 1 and 3 (I141N/G353D[34], Fig. 1c, d). To probe the competition between RNCs and post-translational clients, we performed GroEL:RNC co-sedimentation experiments in the presence of α-lactalbumin, a model client that is constitutively unfolded under reducing conditions[35,36] (Supplementary Fig. 1d). Excess α-lactalbumin did not prevent GroEL co-sedimentation with RNC$_{1-510mut}$, suggesting that GroEL does not necessarily prefer post-translational clients to NCs. We do not exclude that the GroEL double-ring binds α-lactalbumin and the RNC simultaneously.

To understand whether GroEL/ES affects β-gal maturation, we expressed β-gal in a fully-reconstituted in vitro transcription/translation (IVT) system. Supplementing IVTs with GroEL or GroEL/ES did not substantially affect the folding of WT β-gal, but improved the solubility of the I141N/G353D β-gal mutant ~4-fold (Supplementary Fig 1e–h). In summary, GroEL interacts with ribosome-associated NCs, is sensitive to the conformation of the NC, and increases the soluble yield of destabilised nascent β-gal.

### GroEL uses different surfaces to bind ribosomes and nascent chains

GroEL is expected to be flexibly tethered to ribosomes via the partially-folded NC. The resulting assembly is likely to be highly dynamic,

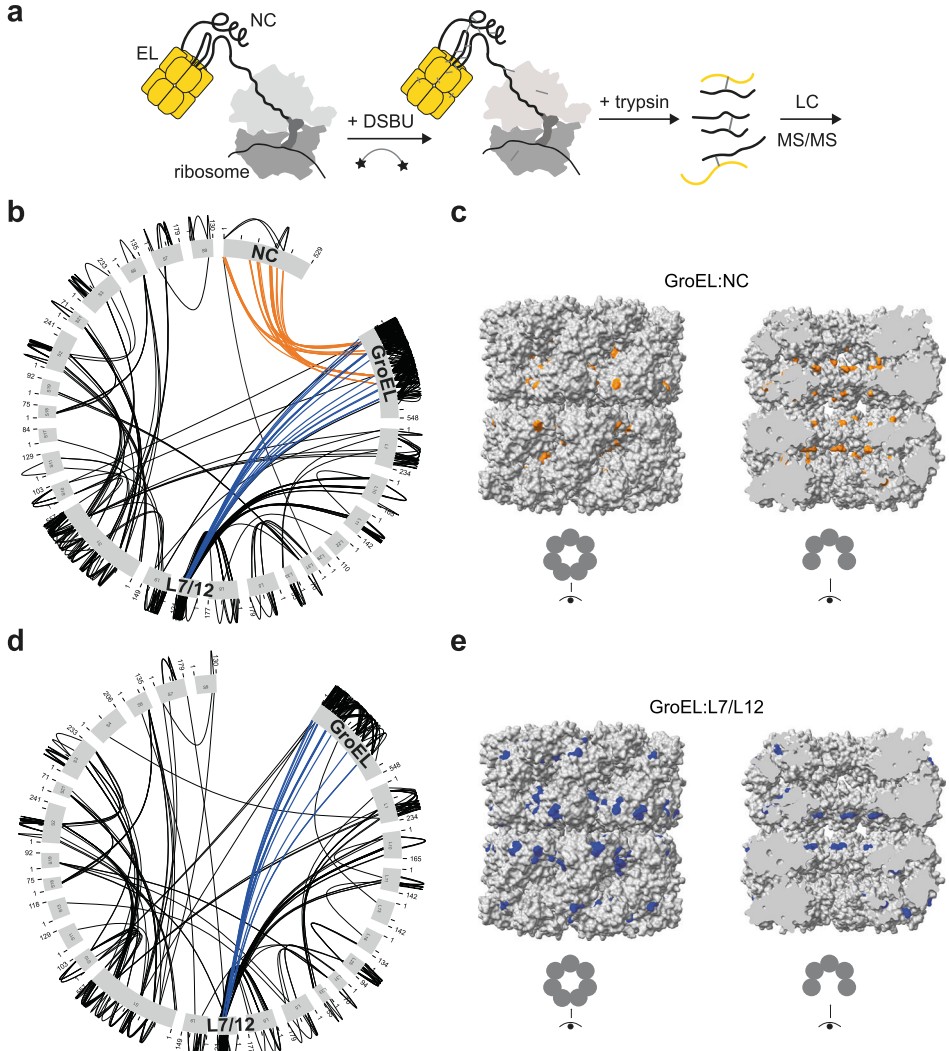

**Fig. 2 | GroEL uses different surfaces to bind ribosomal proteins and nascent chains. a** Crosslinking-mass spectrometry (XL-MS) experiment. **b** GroEL crosslinks extensively to both the NC and ribosomal stalk proteins. Map of crosslinks between GroEL and RNC$_{1\text{-}510mut}$. Crosslinks between GroEL and the NC (orange) or L7/L12 (blue) are highlighted. **c** NCs preferentially crosslink to the inner surface of the GroEL cavity. Position of GroEL residues that crosslink to NC$_{1\text{-}510}$, NC$_{1\text{-}510mut}$, or NC$_{1\text{-}1014}$ are shown in orange on the structure of GroEL (PDB: 5W0S). Left, outer surface. Right, cutaway showing inner surface. **d** GroEL crosslinks to the L7/L12 stalk of empty ribosomes. Map of crosslinks between GroEL and empty 70S ribosomes. Crosslinks between GroEL L7/L12 (blue) are highlighted in blue. **e** L7/L12 preferentially crosslinks to the outer surface of GroEL. Position of GroEL residues that crosslink to L7/L12 in empty ribosomes, NC$_{1\text{-}510}$, NC$_{1\text{-}510mut}$, or NC$_{1\text{-}1014}$ are shown in blue on the structure of GroEL (PDB: 5W0S). Left, outer surface. Right, cutaway showing inner surface. Source data are provided in Supplementary Data 3.

presenting a challenge to structural characterisation. To define the topology of these assemblies, we crosslinked complexes between GroEL and 3 different RNCs using disuccinimidyl dibutyric urea (DSBU) and identified crosslink sites using mass spectrometry (Fig. 2a). DSBU is a homobifunctional crosslinker that preferentially targets lysine residues. GroEL crosslinked extensively to the NC in all cases (Fig. 2b and Supplementary 2a, b). We also identified numerous crosslinks between GroEL and the ribosomal stalk, L7/L12, which coordinates elongation factors during translation[37]. Crosslinking to the stalk did not depend on NC binding, as the same crosslinks formed when GroEL was mixed with empty 70S ribosomes (Fig. 2d). Furthermore, no crosslinks were detected between GroEL and ribosomes stripped of L7/L12 (Supplementary Fig. 2c, d). GroEL therefore engages both NCs and ribosomes, the latter potentially via the stalk complex.

Mapping the crosslink positions on the structure of GroEL revealed a separation of binding interfaces. The majority of crosslinks to the NC mapped to residues inside the GroEL cavity, consistent with a client-like interaction (Fig. 2c). In contrast, L7/L12 preferentially crosslinked to the outer surface of the GroEL (Fig. 2e). Note that

solvent-accessible lysine residues are evenly distributed between the cavity and outer surface of GroEL. GroEL co-sedimentation with RNC$_{1\text{-}510mut}$ was unchanged after removing the L7/L12 stalk, showing that the stalk is not required for stable binding of GroEL to RNCs (Supplementary Fig. 2e).

## Nascent chains bind the apical domains and C-terminal tails of GroEL

We subsequently focused on RNC$_{1\text{-}510mut}$, which formed the most stable complex with GroEL among the tested RNCs (Fig. 1d). To delineate the NC binding surface on GroEL, we measured RNC-induced protection of amide hydrogens using hydrogen/deuterium exchange-mass spectrometry (HDX-MS) (Fig. 3a). Compared to isolated GroEL, GroEL bound to RNC$_{1\text{-}510mut}$ was protected from deuterium exchange at 4 different sites per subunit (Fig. 3b, c). The most protected site was the hydrophobic C-terminal tail of GroEL, which is structurally disordered and protrudes into the cavity (Fig. 3d). A hydrophobic surface on the apical domain (residues 195–214) was also protected, near residues that crosslinked to NC$_{1\text{-}510mut}$ (Fig. 3c, e). This is consistent

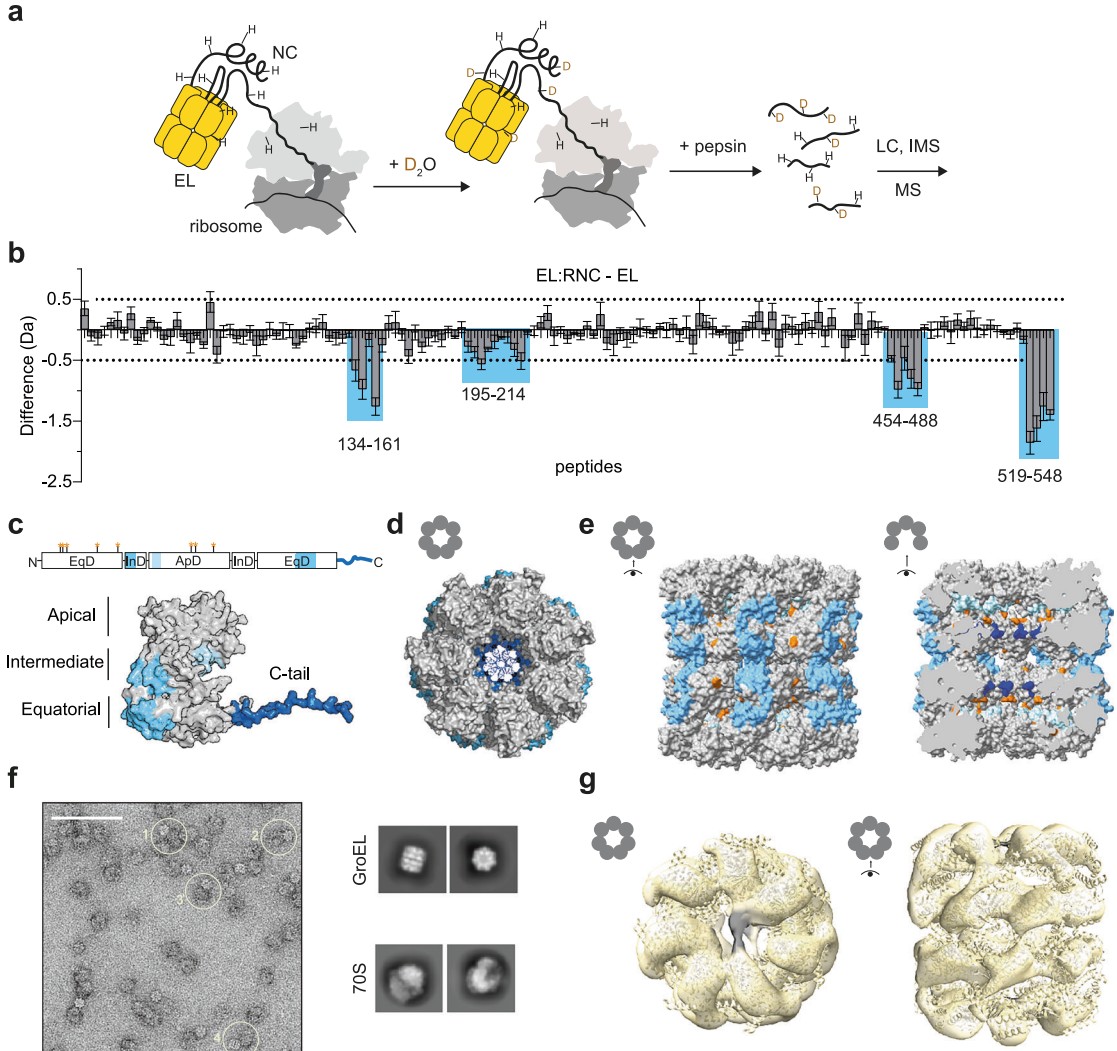

**Fig. 3 | Nascent chains bind the apical domains and C-terminal tails of GroEL.**
**a** Hydrogen/deuterium exchange-mass spectrometry (HDX-MS) experiment.
**b** Protection of GroEL induced by RNC binding. Difference in deuterium uptake after 100 s, between GroEL bound to RNC$_{1-510mut}$ and isolated GroEL. Values are plotted for individual GroEL peptides. Negative values indicate less deuteration of a peptide in RNC-bound GroEL relative to isolated GroEL, and sites with a difference in deuterium uptake >0.5 Da are coloured blue. Data are presented as mean ± SD, n = 3 independent labelling reactions. **c** Sites of RNC$_{1-510mut}$-induced protection from deuterium exchange, mapped onto the domain organisation and predicted structure of monomeric GroEL (AF-P0A6F5-F1). Crosslink sites to NC$_{1-510mut}$ are indicated on the domain schematic with orange asterisks. **d** As in (**c**), shown on a top-view tetradecameric GroEL. The C-tails protrude into the central cavity. **e** As in

(**c**), showing a side-view of tetradecameric GroEL (PDB: 5W0S[84]). Left, outer surface. Right, inner surface. GroEL residues that crosslink to NC$_{1-510mut}$ are shown in orange. Sites with a difference in deuterium uptake >0.5 Da are coloured blue. **f** Visualisation of GroEL:RNC assemblies. Left: Negative stain electron microscopy (nsEM) micrograph of DSBU-crosslinked GroEL:RNC$_{1-510mut}$ assemblies. 500 independent micrographs were collected. The scale bar corresponds to 100 nm. Examples of GroEL molecules positioned near ribosomes are circled (1–4). Right: 2D class averages of GroEL and 70S ribosomes. **g** NC density spans the GroEL cavity at the apical domains. 3D reconstruction of DSBU-crosslinked GroEL:RNC$_{1-510mut}$ (map iv) from nsEM, viewed from the top and side. The structure of GroEL (PDB: 5W0S[84]) is docked into the map, and extra density corresponding to the NC is coloured black. Source data are provided in Supplementary Data 1.

with previous studies showing that residues 199, 201, 203, and 204 are involved in client binding[38], and cryo-EM structures which locate post-translational clients at the apical domains of GroEL[14,39–43]. The two other protected sites (134–161, 454–488) are close together on the external surface of GroEL, near the ATP-binding site (Fig. 3c, e). These peptides are close to residues that crosslink to L7/L12, suggesting that the protection in this region might result from interactions with the ribosome rather than NC. We do not exclude the possibility that the protection near the ATP-binding pocket is allosterically induced by client binding inside the cavity.

Negative stain electron microscopy (nsEM) of GroEL:RNC complexes showed particles identifiable as GroEL or ribosomes, and instances of GroEL positioned close to ribosomes (Fig. 3f and Supplementary Fig. 3a). Single-particle analysis resulted in a low-resolution

reconstruction of ribosome-associated GroEL, and fitting the solved structure of GroEL into the map revealed additional density contacting the apical domains at the cavity opening (Fig. 3g and Supplementary Fig. 3b, c). This density was poorly resolved in an untreated sample (Supplementary Fig. 3b), but clearly spanned the cavity in maps from DSBU-crosslinked samples (Fig. 3g and Supplementary Fig. 3c).

In summary, our XL-MS, HDX-MS and nsEM analyses indicate that the NC binds near the top of the GroEL cavity, as previously observed for chemically denatured client proteins. Moreover, we provide evidence that the disordered C-terminal tails of GroEL directly contact the NC. Although dispensable in vivo[44], the C-tails have previously been shown to contribute to client capture and folding[42,45–48], and recent work suggests that they are required for optimal chaperonin function[49].

## GroES can be accommodated on RNCs bound by GroEL

During chaperonin-mediated protein folding, ATP binding to GroEL induces a conformational change that primes the chaperone for binding GroES, resulting in client encapsulation in the cis cavity (Fig. 1a). As RNCs (>2.5 MDa) substantially exceed the size limit of the GroEL/ES chamber (~70 kDa,[8]), we sought to understand whether the same mechanism applies co-translation. To address this question, we used nucleotides in combination with metal salts to stabilise specific intermediates in the GroEL/ES reaction cycle (Fig. 1a). ADP/BeF$_x$ mimics ATP but is non-hydrolysable, and stabilises the asymmetric complex with GroEL capped on one end by GroES[50]. Adding ATP and BeF$_x$ allows ATP binding and hydrolysis prior to trapping by BeF$_x$; the resulting complex contains ADP/BeF$_x$ in both rings and is symmetrically capped by GroES[50].

We first tested the nucleotide dependence of GroEL binding to RNCs in the absence of GroES. Addition of either ADP, ATP, ADP/BeF$_x$ or ATP/BeF$_x$ reduced the amount of GroEL that co-sedimented with RNC$_{1-510mut}$ (Supplementary Fig. 4a). The conformational changes triggered by nucleotide binding therefore weaken, but do not completely disrupt, the interaction between NCs and GroEL.

GroES can bind either ring of GroEL, resulting in a mixture of single- and double-capped complexes. To generate homogenous assemblies, we supplemented the reactions with ATP/BeF$_x$, established to result in symmetric EL:ES$_2$ complexes with both cavities bound by GroES[50] (Fig. 4a). Pre-forming the symmetrically-closed EL:ES$_2$ complex prevented either GroEL or GroES from co-sedimenting with RNCs, demonstrating that NC binding requires access to the chaperonin cavity (Fig. 4b). In contrast, GroEL/ES co-sedimented with the RNC if GroEL was allowed to bind prior to addition of GroES and ATP/BeF$_x$ (Fig. 4b and Supplementary Fig. 4a, b). The amount of co-sedimenting GroEL decreased only slightly in the presence of GroES and ATP/BeF$_x$, from 9.8 ± 0.9 to 6.7 ± 1.5 GroEL subunits per ribosome (Fig. 4c and Supplementary Data 2).

## GroEL/ES partially encapsulates ribosome-bound nascent chains

Since we were able to stabilise GroES on GroEL:RNC complexes (Fig. 4b), we next asked whether the NC in these complexes is encapsulated in the cis cavity underneath GroES, or bound to the GroES-free trans ring. We repeated the GroEL/ES:RNC co-sedimentation experiments using a single-ring variant of GroEL with only one folding chamber (SR1[51]) (Supplementary Fig. 4c). In the presence of ATP/BeF$_x$, we found that similar amounts of GroES co-sedimented with RNC$_{1-510mut}$ bound to either wild-type GroEL or SR1, suggesting that GroES can bind the cis cavity of RNC-occupied GroEL.

Next, we used XL-MS to probe the architecture of the ATP/BeF$_x$-stabilised GroEL:ES:RNC complex. Similar to GroEL:RNC complexes, both GroEL and GroES crosslinked to the NC and ribosomal stalk (Fig. 5a and Supplementary Fig. 5a). Residues that crosslinked to L7/L12 mapped exclusively to the outside of GroEL/ES, while residues that crosslinked to the NC were biased to the inside of the EL/ES cavity (Fig. 5b and Supplementary Fig. 5b). The latter included four residues (226, 360, 364, 371) that did not crosslink to NCs in GroEL:RNC complexes (Fig. 2b and Supplementary Fig. 2a, b). Two of these residues (364, 371) had crosslinked to L7/L12 in the EL:RNC complexes (Supplementary Data 3). The other two (226, 360) were previously shown to interact with client proteins in the closed EL:ES$_2$ complex[42]. Mapping these residues onto the GroEL and EL:ES$_2$ structures showed that they were positioned inside the GroEL cavity in the EL:ES$_2$ complex, but not apo-GroEL, explaining the difference in crosslinking patterns (Supplementary Fig. 5c).

The position of the nascent chain inside the GroEL/ES cavity was further supported by nsEM of the EL:ES$_2$:RNC complex. Approximately 90% of GroEL/ES complexes were double-capped EL:ES$_2$ (Fig. 5c and Supplementary Fig. 5d), and analysis of these species resulted in two

3D reconstructions which revealed additional density inside one or both EL/ES cavities (Fig. 5d and Supplementary Fig. 5e).

As an orthogonal test of NC encapsulation, we treated RNC$_{1-510mut}$ complexes with proteinase K (Fig. 6a, b and Supplementary Fig. 6a). Addition of either GroEL or GroEL/ES/ATP/BeF$_x$ slowed proteolysis of the full-length tRNA-NC species. Moreover, EL:ES$_2$ uniquely protected two ~60 kDa proteolysis intermediates. These were truncated near the C-terminus, as they had lost the peptidyl tRNA (Fig. 6a) but reacted with an antibody against the N-terminus of β-gal (Supplementary Fig. 6a). The same proteolysis intermediates were observed when EL:ES$_2$:RNC complexes were isolated from excess unbound chaperone prior to protease treatment (Supplementary Fig. 6b,c). This phenomenon was not unique to RNC$_{1-510mut}$, as EL:ES$_2$ protected a similarly sized intermediate generated during proteolysis of RNC$_{1-687}$ (Fig. 6b and Supplementary Fig. 6d). These data indicate that GroEL:ES$_2$ can encapsulate a ~60 kDa NC fragment prior to NC release from the ribosome, consistent with the estimated size limit of the GroEL/ES cavity[8].

In summary, our co-sedimentation, XL-MS, nsEM, and limited proteolysis data demonstrate that ribosome-tethered NCs are partially encapsulated by GroEL/ES.

## GroEL binding destabilises the NC prior to encapsulation

We next sought to understand how GroEL binding affects the conformation of the NC. We analysed GroEL:RNC and EL:ES$_2$:RNC complexes using HDX-MS, and compared deuterium uptake to isolated RNC$_{1-510mut}$. Peptide coverage of the nascent chain was limited to ~53% by the complexity of the sample containing both ribosome- and GroEL-derived peptides (Supplementary Fig. 7a, b). We found that the N-terminal part of the NC (residues 8-83) was deprotected by 0.5–1.5 Da upon GroEL binding, indicative of local conformational destabilisation (Fig. 7a). Comparison of NC$_{1-510mut}$ to native full-length β-gal showed that residues 8-24 were highly deprotected (>4 Da) in the NC, while residues 25–83 showed similar deuterium uptake to the native state (Δ ~0.3–0.7 Da, Fig. 7b). GroEL binding therefore destabilises a native-like region in the NC, and further destabilises an already partially unfolded segment.

Since GroEL selectively destabilised the N-terminal region of the NC, we next asked whether this region specifically binds the chaperone. We found that the N-terminus is not required for binding, as deletion of the first 24 or 81 residues did not disrupt the GroEL:RNC interaction (Supplementary Fig. 7c). Instead of uniquely binding the N-terminus, our data argue that GroEL binds multiple sites across the NC and prevents the N-terminal region from making stabilising intramolecular contacts. This is consistent with our XL-MS analysis, which identified crosslinks to GroEL throughout the NC (Fig. 7a).

GroEL-induced NC destabilisation was reversed upon encapsulation in EL:ES$_2$ (Fig. 7a). No detected peptides differed significantly in exchange compared to the unbound NC, indicating that the NC recovered its original fold. The NC therefore interacts differently with GroEL compared to EL:ES$_2$, consistent with the burial of hydrophobic sites on GroEL upon GroES binding[2]. This difference was also apparent when we analysed deuterium uptake of GroEL itself (Supplementary Fig. 7d). The client binding sites in the apical domains of apo GroEL (195–214, Fig. 3) were unaffected by NC binding to EL:ES$_2$, consistent with burial of these sites at the interface with GroES[8]. Furthermore, the flexible C-termini were not protected by RNC binding to EL:ES$_2$, indicative of reduced involvement in client binding compared to apo-GroEL.

## GroEL competes with TF and DnaK

We previously showed that β-gal RNCs bind both DnaK and TF, with TF outcompeting DnaK at ribosome-proximal sites[31]. We therefore asked whether GroEL binding to RNCs is influenced by TF and DnaK (Fig. 8a).

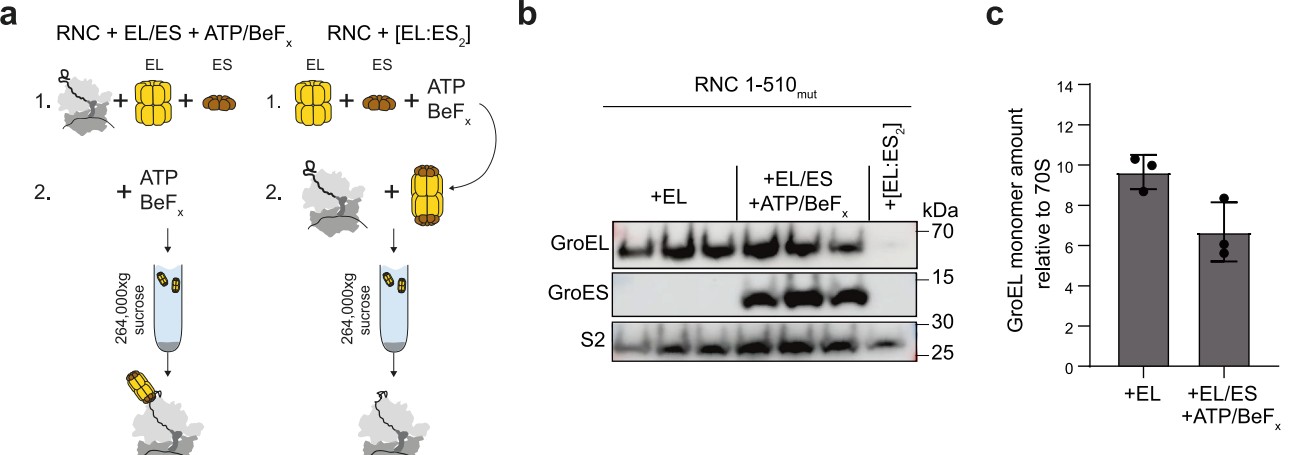

**Fig. 4 | GroES can be accommodated on RNCs bound by GroEL. a** GroEL/ES:RNC co-sedimentation assay. Two different orders of addition were followed. Left: RNCs were incubated with GroEL/ES before addition of ATP/BeFx. Right: GroEL/ES was pre-mixed with ATP/BeFx before incubation with RNCs. **b** GroES co-sediments with a GroEL:RNC complex. Immunoblot of the resuspended ribosomal pellets from triplicate co-sedimentation assays described in (**a**), probed using antibodies against GroEL, GroES and the ribosomal protein S2. **c** Stoichiometry of GroEL:RNC complexes. Mean intensity-based absolute quantification (iBAQ) values for GroEL in resuspended ribosomal pellets from co-sedimentation assays described in (**a**). Values are normalised to the average iBAQ of all ribosomal proteins in each sample. Data are presented as mean ± SD, $n = 3$ independent co-sedimentation assays. Source data are provided as a Source Data file and in Supplementary Data 2.

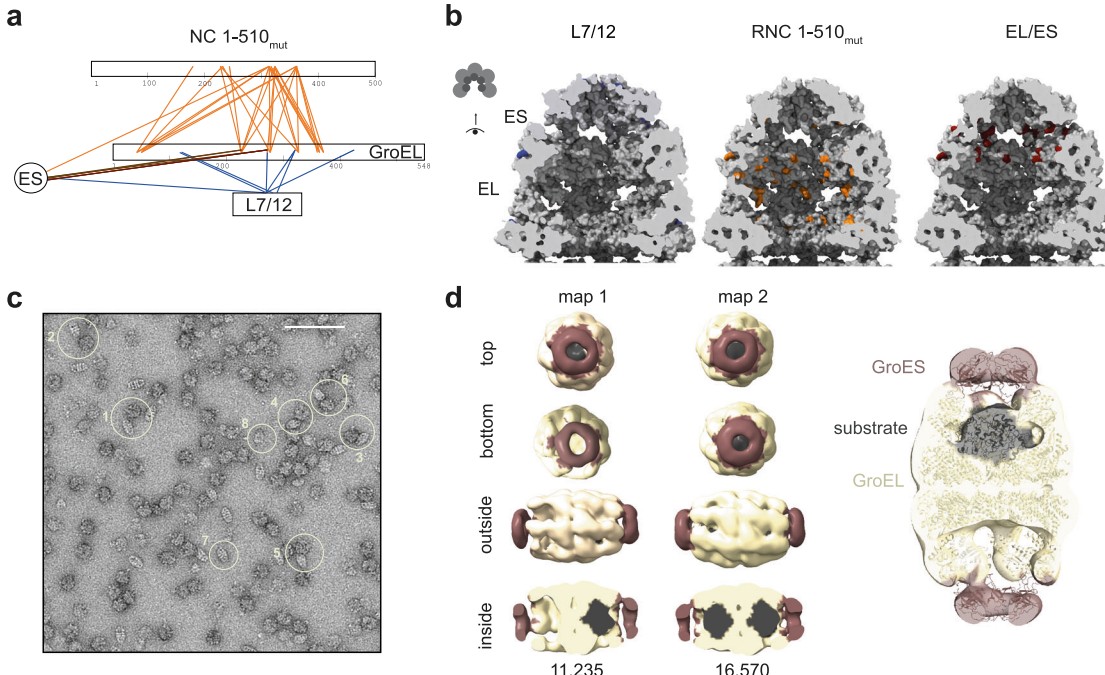

**Fig. 5 | Nascent chains occupy the central cavity of GroEL/ES. a** Map of crosslinks between GroEL/ES and RNC$_{1-510mut}$ in the EL:ES$_2$:RNC complex. Only crosslinks between GroEL/ES and L7/L12 (blue), GroEL and GroES (brown), and GroEL/ES and the NC (orange) are shown. **b** NC and ribosome stalk crosslink to the inner and outer surface of GroEL/ES, respectively. Crosslink sites are mapped onto the structures of ATP/BeFx-stabilised EL:ES$_2$ (PDB:7VWX[85]), cut away to show the inner surface of the GroEL/ES cavity. Residues are separated according to whether they crosslink to L7/L12 (blue), the NC (orange), or connect GroEL and GroES (brown). **c** Visualisation of GroEL:ES$_2$:RNC assemblies. Micrograph from nsEM of GroEL:ES$_2$:RNC$_{1-510mut}$ complex. The scale bar corresponds to 100 nm. Examples of GroEL:ES$_2$ complexes positioned near ribosomes (1-6), isolated double-capped EL:ES$_2$ (7), or single-capped EL:ES$_1$ (8) complexes are circled. 500 independent micrographs were collected. **d** NC density in the GroEL/ES cavity. Left: 3D reconstructions of GroEL:ES/RNC complexes from nsEM. Two classes can be distinguished, with additional density (black) in one or both chambers. The number of particles contributing to each reconstruction is given at the bottom. Right: The structure of GroEL/ES with encapsulated Rubisco (PDB: 7VWX[85]) docked into map 1. Source data are provided in Supplementary Data 3.

Since RNC$_{1-510}$ is a poor TF substrate[31], we prepared RNC$_{1-180}$, RNC$_{333-510}$ and RNC$_{1-1014}$ which bind both TF and GroEL when added individually (Fig. 8b, c). For the shorter NCs, GroEL binding was completely outcompeted by equimolar TF (Fig. 8b). In contrast, GroEL or GroEL/ES binding to RNC$_{1-1014}$ was not affected by, nor did it affect, TF binding (Fig. 8c and Supplementary Fig 8a). This suggests that TF and GroEL can occupy independent sites on NCs, with TF preferred at the ribosome-proximal site. Simultaneous addition of GroEL and DnaK led to reduced levels of both chaperones on RNC$_{1-510}$ or RNC$_{1-510mut}$ (Fig. 8d). The same behaviour was observed in the presence of GroES

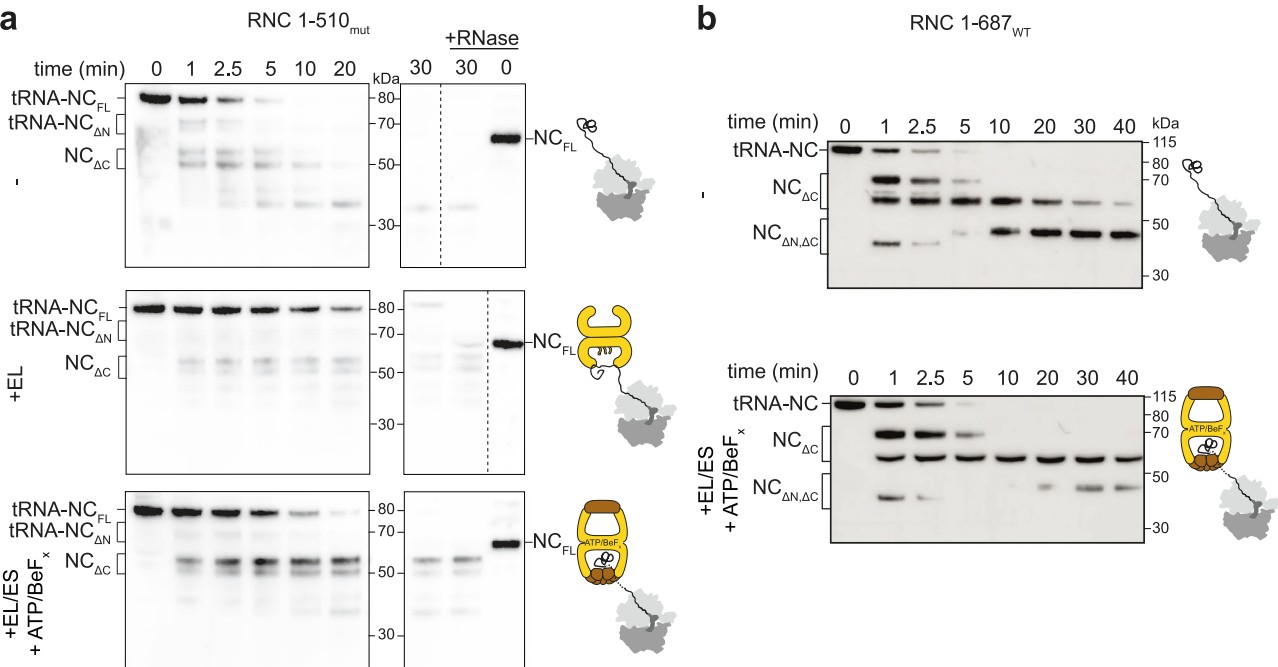

**Fig. 6 | GroEL/ES protects ~ 60 kDa fragments of nascent β-gal from limited proteolysis. a** Limited proteolysis of RNC$_{1-510mut}$. NC$_{1-510mut}$ was detected by immunoblotting using an antibody raised against full-length β-gal, at different times following addition of proteinase K. Limited proteolysis was performed on isolated RNC$_{1-510mut}$ (top), RNC$_{1-510mut}$ incubated with GroEL (middle) and RNC$_{1-510mut}$ incubated with GroEL and GroES followed by addition of ATP/BeF$_x$ (bottom). Bands corresponding to the tRNA-bound full-length NC (tRNA-NC$_{FL}$) and degradation intermediates truncated at the N-terminus (tRNA-NC$_{ΔN}$) or C-terminus (NC$_{ΔC}$) are highlighted. Samples in the last two lanes were additionally treated with RNase to identify the NC species covalently linked to tRNA. The experiment was repeated twice with similar results. **b** Limited proteolysis of RNC$_{1-687}$, performed as in (A), for isolated RNC$_{1-687}$ (top) and RNC$_{1-687}$ incubated with GroEL and GroES followed by addition of ATP/BeF$_x$ (bottom). The experiment was repeated twice with similar results. Source data are provided as a Source Data file.

and ATP/BeF$_x$ (Supplementary Fig. 8b). GroEL and DnaK therefore act in a mutually antagonistic manner at the ribosome.

## Discussion

Our data provide a mechanistic underpinning for chaperonin function during cotranslational folding (Fig. 8e). Despite the proximity of the ribosome, nascent polypeptides access sites deep in the GroEL cavity, including the apical domains and C-terminal tails. This interaction locally unfolds the NC, which then refolds following displacement into the cis cavity by GroES. GroEL recognises NCs that are poor TF substrates, and competes with DnaJ/K.

We show that GroEL/ES unfolds then partially encapsulates nascent polypeptides. GroEL-induced client expansion was previously observed, and is speculated to result from multivalent contacts with the chaperone cavity[14,48,52–54]. Alternatively, binding of the bulky GroEL molecule may cause partial unfolding of distal segments of the NC via entropic pulling[55]. In the context of cotranslational folding, local unfolding may resolve misfolded states prior to refolding in the GroEL/ES cage. Neither TF nor DnaK unfold β-gal NCs[31], suggesting that this may be a specialised function of GroEL. β-gal assembles cotranslationally[56], but assembly initiates only upon emergence of the first 4 domains from the ribosome[57]. For oligomeric proteins such as β-gal, GroEL/ES may play a role in shielding N-terminal domains exposing unsatisfied assembly interfaces, prior to the onset of cotranslational assembly.

The encapsulation of nascent chains is significant, as the chaperonin cavity is a unique folding environment which can modulate the energy landscape of protein folding. Physical confinement, the net negative charge of the cavity wall, and interactions with the disordered C-termini have all been shown to contribute to accelerated folding inside GroEL/ES[2]. Furthermore, encapsulated clients are insulated from intermolecular interactions, allowing folding at effectively infinite dilution. Our data show that nascent proteins can benefit from folding within the chaperonin cage before their synthesis is complete. Early GroEL/ES engagement might increase the efficiency of chaperonin-mediated protein folding in some cases, by bypassing the Hsp70 system[19]. The relative importance of co- versus post-translational chaperonin action in vivo remains to be determined.

Cotranslational encapsulation implies that the NC protrudes from the closed chaperonin cavity to remain connected to the ribosome. This represents a topological problem, since GroES is thought to bind en bloc to GroEL and seal the cavity. One possibility is that GroES binds asymmetrically to GroEL, leaving a gap through which the NC can thread. Indeed, previous work has suggested that clients might pass through a space at the GroEL/ES interface[58,59]. Furthermore, a recent structure of GroEL bound to Rubisco and ADP/BeF$_3$ showed a subset of GroEL-domains in a GroES-binding state, suggesting that partial docking of GroES on GroEL is possible[42]. Understanding exactly how encapsulation is achieved will require structural resolution of the nascent polypeptide, a significant challenge due to the conformational heterogeneity of partially-folded clients. Our NsEM data revealed client density in both rings of EL:ES$_2$ complexes, suggesting that GroEL might bind co- and posttranslational clients simultaneously, or engage two NCs at once. The latter scenario would be facilitated by the clustering of ribosomes in polysomes.

Partial encapsulation may also be relevant post-translation. A fraction of obligate GroEL/ES clients are too large (>70 kDa) to be completely encapsulated[25,60–63], and many large proteins are efficiently refolded by GroEL/ES[64,65]. This has been proposed to occur without cis encapsulation, via iterative binding and release from the trans ring of GroEL[61]. Our data suggest the alternative possibility that some large GroEL clients fold via domain-wise encapsulation in the cis cavity. Indeed, a segmental encapsulation mechanism has previously been

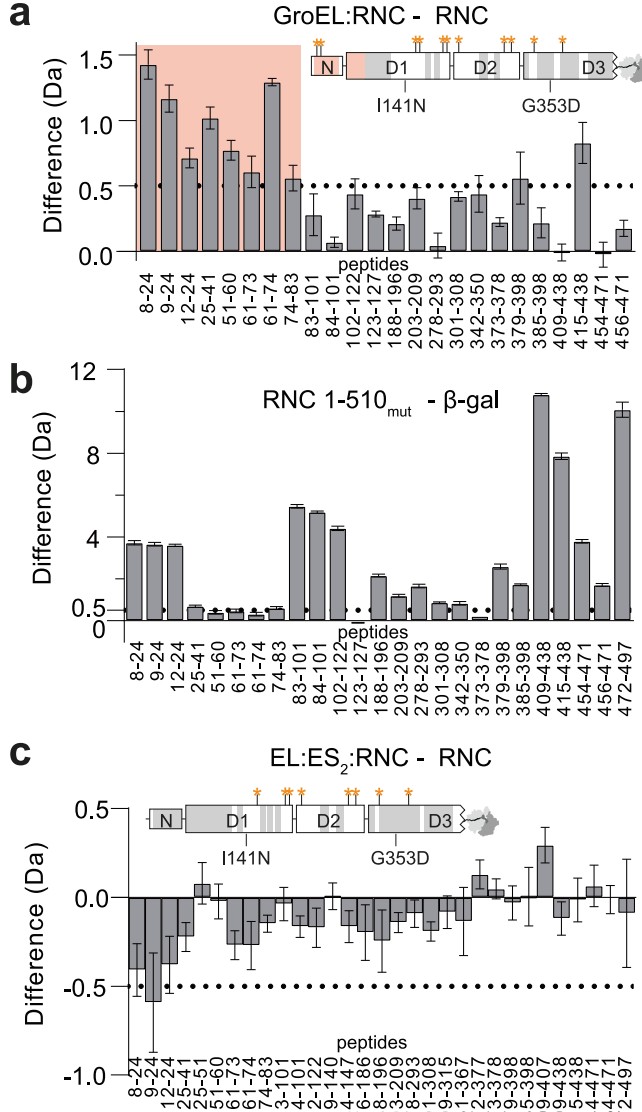

**Fig. 7 | GroEL locally destabilises the NC prior to encapsulation in GroEL:ES$_2$.**
**a** Deprotection of NC$_{1\text{-}510\text{mut}}$ upon binding GroEL. Difference in deuterium uptake after 100 s, between GroEL-bound RNC$_{1\text{-}510\text{mut}}$ and isolated RNC$_{1\text{-}510\text{mut}}$. Values are plotted for individual peptides covering the β-gal NC. Positive values indicate more deuteration of a peptide in the GroEL-bound RNC relative to the isolated RNC. The data are also mapped onto a schematic domain diagram of NC$_{1\text{-}510\text{mut}}$. Sites with increased deuterium uptake are shaded in red, sites with unchanged deuterium uptake are shaded in grey, and sites on the NC that crosslinked to GroEL are indicated using orange asterisks. Unshaded regions are not covered by peptides. Data are presented as mean ± SD, $n = 3$ independent labelling reactions.
**b** Conformational dynamics of NC$_{1\text{-}510\text{mut}}$ compared to native β-gal. Difference in deuterium uptake after 100 s, between isolated RNC$_{1\text{-}510\text{mut}}$ and native full-length β-gal. Positive values indicate more deuteration of a peptide in the RNC relative to native β-gal. Data are presented as mean ± SD, $n = 3$ independent labelling reactions. **c** No deprotection of NC encapsulated by GroEL/ES. As in (**a**), except isolated RNC$_{1\text{-}510\text{mut}}$ is compared to the complex with GroEL/ES/ATP/BeF$_x$. Data are presented as mean ± SD, $n = 3$ independent labelling reactions. Source data are provided in Supplementary Data 1.

demonstrated for the eukaryotic chaperonin TRiC[66]. Whether TRiC can also encapsulate ribosome-tethered domains is not clear.

We show that GroEL weakly interacts with ribosomes independent of the NC. This occurs via the outer surface of the chaperonin cavity, via sites distinct from those that bind clients. Although XL-MS suggests that GroEL binds directly to the ribosomal L7/L12 stalk, we do not

exclude that a different interface is involved. For example, GroEL binding to ribosomal RNA would not be detected by DSBU crosslinking.

Whether GroEL binding to the ribosome influences the cotranslational activity of the chaperonin remains to be determined. Weak interactions with the ribosome may promote NC capture by increasing the local concentration of GroEL, or contribute avidity for NCs that are poor GroEL clients. Ribosome binding may also increase the "processivity" of GroEL/ES, by keeping GroEL close to the ribosome between ATP-driven cycles of NC binding and release.

The L7/L12 stalk is involved in recruiting elongation factors to the ribosome[37,67]. If GroEL does bind directly to this site, it raises the intriguing possibility that GroEL may slow translation elongation during chaperonin-assisted cotranslational folding.

Although GroEL may in principle recognise many different nascent polypeptides, competition with other cytosolic chaperones restricts GroEL to a subset of NCs in vivo. Indeed, simultaneous deletion of DnaK/DnaJ and TF was found to increase the level of GroEL at ribosomes ~10-fold[22]. We show that TF efficiently outcompetes GroEL for binding to short NCs, but both chaperones can be accommodated on longer NCs. Longer NCs are also clients of DnaK/J[31], resulting in competition between GroEL and DnaK[23]. GroEL and DnaK/J may directly compete for similar binding sites on the NC, or the action of one chaperone system could change the conformation of the NC so that it is no longer recognised by the other. Considering that GroEL strongly prefers conformationally destabilised NCs, it may be recruited to nascent domains that fail to fold while they are close to the ribosome surface, and thus persistently expose hydrophobic segments beyond the reach of TF.

## Methods

### DNA vectors and cloning

The pET28-based plasmid encoding full-length β-galactosidase and the pET21-based plasmids encoding muGFP-tagged[68] RNC complexes stalled with SecM-based WWWPRIRGPP stalling sequence[69] were cloned in our previous study[31]. *E. coli* DnaK and DnaJ were expressed without any tags from a pET11d-based vector. His$_{6x}$-tagged TF was expressed from a ProEX backbone. GroEL and GroES were expressed from pET-17b-based vectors (Novagen). Additional deletions and point mutations were introduced using site-directed mutagenesis with Q5 or Phusion polymerases (NEB). All constructs used in this study (Supplementary Data 4) were verified by sequencing. The plasmid encoding GroEL-SR1 was a kind gift from F.U. Hartl.

### RNC buffers

RNC low-salt buffer contained 50 mM HEPES-NaOH pH 7.5, 12 mM Mg(OAc)$_2$, 100 mM KOAc, 1 mM DTT and 8 U/mL RiboLock RNase inhibitor (ThermoScientific). RNC high-salt sucrose cushion contained 35% sucrose, 50 mM HEPES-NaOH pH 7.5, 12 mM Mg(OAc)$_2$, 1 M KOAc, 1 mM DTT, 8 U/mL RiboLock RNase inhibitor and 0.2x Halt Protease Inhibitor Cocktail (ThermoScientific). RNC low-salt sucrose cushion contained 35% sucrose, 50 mM HEPES-NaOH pH 7.5, 12 mM Mg(OAc)$_2$, 100 mM KOAc, 1 mM DTT, 8 U/mL RiboLock RNase inhibitor and 0.2x Halt Protease Inhibitor Cocktail.

### Protein purification

RNCs, Full-length β-galactosidase, DnaK, DnaJ and Trigger factor were expressed and purified as described previously[31]. Sequences of purified proteins are listed in Supplementary Data 4. RNC$_{1\text{-}510\text{mut}}$ contained two previously characterised destabilising mutations: I141N and G353D[34].

In this study we used mutated GroEL (V381W, A384W, V387W) and GroES (L49W), which introduced tryptophan residues into external flexible loops not known to have any role in the activity and function of the chaperone (Supplementary Fig. 1a). The mutations facilitated

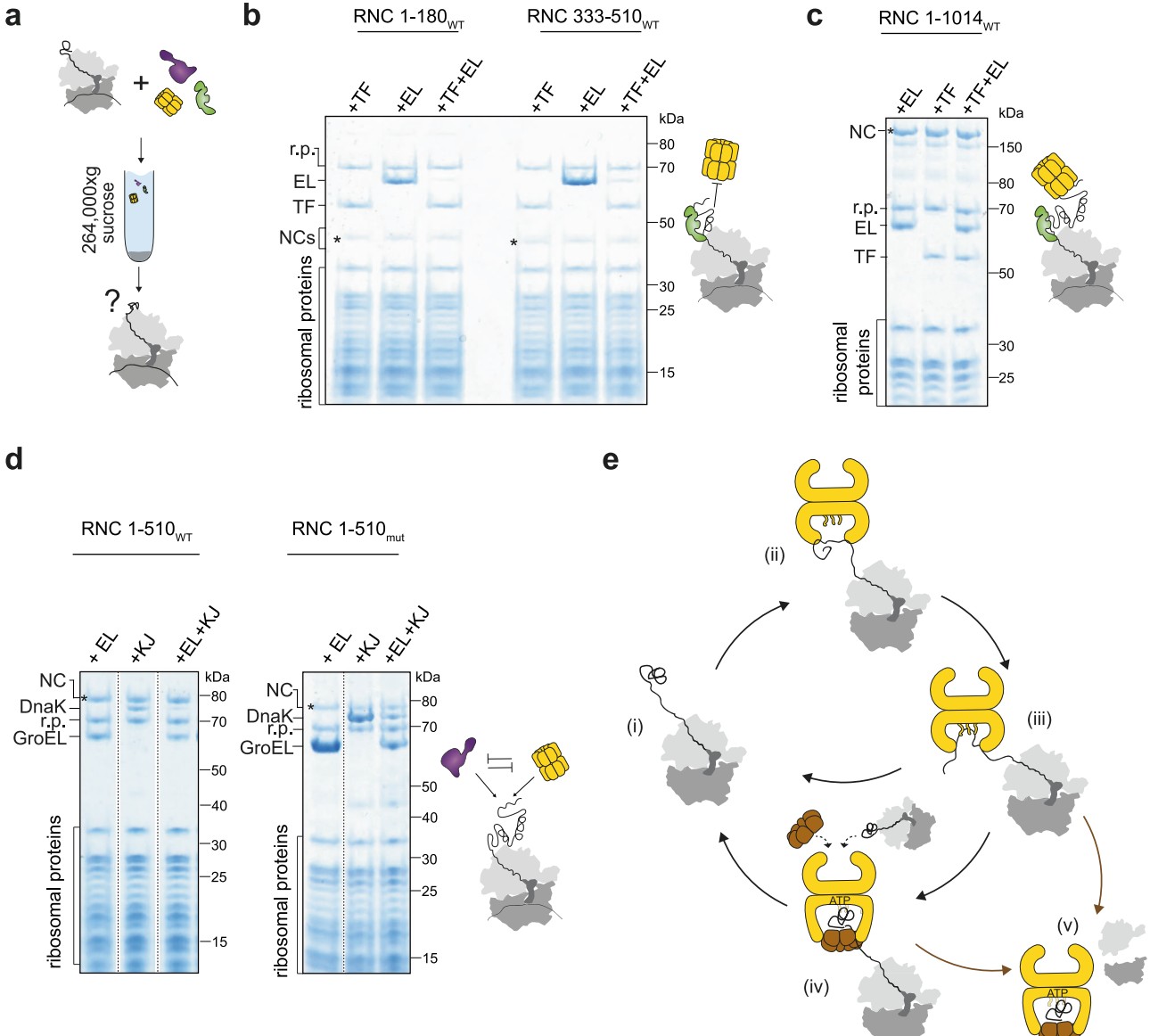

**Fig. 8 | Coordination of GroEL with Trigger factor and DnaK. a** Multi-chaperone co-sedimentation assay. RNCs were incubated with GroEL and either TF or DnaK/DnaJ/ATP before centrifugation through a 35% sucrose cushion to separate the ribosomal fraction (pellet) from unbound chaperones (supernatant). **b** TF outcompetes GroEL at short NCs. Coomassie-stained SDS-PAGE of the resuspended ribosomal pellet from the co-sedimentation assay. $RNC_{1-180}$ or $RNC_{333-510}$ were incubated with GroEL (+EL), Trigger factor (+TF), or both chaperones together (+EL + TF). Bands corresponding to the NC (*), TF, GroEL, and ribosomal proteins (r.p.) are indicated. The experiment was repeated twice with similar results. **c** TF and GroEL do not compete for binding long NCs. Chaperone co-sedimentation with $RNC_{1-1014}$ was tested as in (b). The experiment was repeated twice with similar results. **d** GroEL and DnaK compete for binding RNCs. Coomassie-stained SDS-PAGE of the resuspended ribosomal pellet from the co-sedimentation assay as in

(a). $RNC_{1-510WT}$ or $RNC_{1-510mut}$ were incubated either with GroEL (+EL), DnaK/DnaJ/ATP ( + KJ) or both (+EL + KJ). Bands corresponding to the NC (*), DnaK, GroEL, and ribosomal proteins (r.p.) are indicated. The experiment was repeated twice with similar results. **e** Model for the cotranslational action of GroEL/ES. Poorly folded NCs (i) are captured by the apical domains and C-terminal tails of GroEL or EL:ES$_1$, resulting in further unfolding (ii and iii). GroEL may dissociate, or the cis ring may be capped by GroES to partially encapsulate the ribosome-bound NC (iv). In this state it is possible that the trans ring may capture a second NC or post-translational client. The NC may then fold enough to exclude GroEL/ES, or remain encapsulated and complete folding in the chaperonin cavity post-translation (v). Some substrates may only become encapsulated after translation termination (arrow from iii to v). Source data are provided as a Source Data file.

accurate quantification of GroEL and GroES, which lack endogenous tryptophan residues, but did not affect their activity as confirmed by an eGFP refolding assay (Supplementary Fig. 1b). GroEL was overexpressed in *E. coli* Δ*lacZ* cells[70] and induced at 37 °C with 1 mM IPTG overnight. Cells were harvested (4000 g, 20 min) and resuspended in Buffer L (30 mM Tris-HCl pH 7.5, 30 mM NaCl, 1 mM EDTA, 1 mM DTT) with lysozyme, universal nuclease (Thermo Fisher), and cOmplete EDTA-free protease inhibitor cocktail. Following lysis by sonication, the soluble fraction was isolated through centrifugation (50,000 g, 1 h,

4 °C) and loaded onto a self-packed DEAE column equilibrated in Buffer L. Peak fractions eluted with an NaCl gradient in Buffer L were combined. buffer exchanged into Buffer L, and loaded onto a heparin column equilibrated in Buffer L. Peak fractions eluted with an NaCl gradient were combined, diluted 3-fold with Buffer L, and loaded onto Resource Q column preequilibrated in Buffer L. Peak fractions eluted with an NaCl gradient were combined, diluted 3-fold with Buffer L, and loaded onto a heparin column preequilibrated in Buffer L. Peak fractions eluted with an NaCl gradient were combined, concentrated, and

purified further using a Superose 6 size-exclusion column. Protein concentration was determined by absorbance using an extinction coefficient of $21{,}430 \, M^{-1} \, cm^{-1}$ and a molecular weight of 57.6 kDa. Single-ring variant of GroEL (R452E/E461A/S463A/V464A[51]) was purified following the same procedure.

GroES (L49W) with an N-terminal double StrepII-tag was overexpressed in *E. coli* BL21(DE3) cells (NEB) and induced at 37 °C with 1 mM IPTG overnight. Cells were harvested (4000 g, 20 min) and resuspended in Buffer L with lysozyme, universal nuclease (Thermo Fisher), and cOmplete EDTA-fee protease inhibitor cocktail. Following lysis by sonication, the soluble fraction was isolated through centrifugation (50,000 g, 1 h, 4 °C) and loaded onto a StrepTactin-II column. Peak fractions eluted with 2 mM d-desthiobiotin in Buffer L. The affinity tag was cleaved by overnight incubation with GST-3C protease. The sample was concentrated and purified further on a Superdex 200 column. Excessive 3 C protease was removed using a GSTrap column. Protein concentration was determined by absorbance using an extinction coefficient of $6690 \, M^{-1} \, cm^{-1}$ and a molecular weight of 10.5 kDa.

### GFP refolding assay

The eGFP refolding assay protocol was modified from previously described protocol[71]. Purified eGFP[72] was denatured at 1.25 µM in 30 mM HCl for 1 hour at 25 °C. The denatured eGFP was then diluted 100-fold into either 30 mM HCl (denatured), refolding buffer (50 mM MOPS pH 7.0, 100 mM KCl, 5 mM DTT, 0.0125% TWEEN-20, 10 mM Mg-acetate), or refolding buffer with 4 µM GroEL (WT or V381W, A384W, V387W) and 8 µM GroES (WT or L49W). The refolding was monitored in a time-course measurement on a ClarioStar Plus plate reader (BMG Labtech) at 25 °C after the addition of ATP to a final concentration of 2 mM. The samples were excited at 470 nm (15 nm bandwidth) and the emission was recorded at 515 nm (20 nm bandwidth). To plot the fluorescence recovery, detected fluorescence signal was normalised to the fluorescence signal of native eGFP in refolding buffer with 2 mM ATP. Fluorescence recovery after 20 min of refolding (when refolding reached a plateau) was plotted.

### Depletion of L7/L12 from ribosomes

*E. coli* 70S ribosomes or RNCs were depleted of the ribosomal stalk L7/L12 proteins using an adapted protocol based on $NH_4Cl$/ethanol treatment[73]. Complete 70S ribosomes (33 µL, 13.3 µM, NEB) were mixed and incubated (4 °C, 10 min) with 450 µL of ice-cold stripping buffer (20 mM Tris-HCl, pH 7.5, 0.6 M $NH_4Cl$, 20 mM $MgCl_2$, 5 mM β-mercaptoethanol, RiboLock RNase inhibitor). Subsequently, 250 µL of ice-cold ethanol was added to the sample, mixed and incubated (4 °C, 10 min) followed by a second addition of 250 µL of ice-cold ethanol and incubation (4 °C, 5 min). Sample was then layered over a low-salt sucrose cushion and centrifuged (4 °C, 264,000 x g, 2 h) to selectively pellet ribosomes. The pellet containing 70S ribosomes or RNCs depleted of the ribosomal stalk L7/L12 proteins was resuspended in RNC low-salt buffer and the depletion was confirmed by immunoblotting. For RNCs, integrity of the NC linked to peptidyl-tRNA was confirmed by SDS-PAGE and Coomassie staining.

### Proteinase K assay

RNC was diluted to 0.25–3 µM in RNC low-salt buffer and incubated (30 min, 30 °C) either alone, with 5-fold molar excess GroEL 14-mer, or with 5-fold molar excess GroEL 14-mer and 10–20-fold molar excess GroES 7-mer. The sample with GroES was subsequently supplemented with 1 mM ATP, 1 mM $BeSO_4$ and 10 mM NaF and incubated for further 10 min at 30 °C. Samples were then cooled down to 4 °C. Optionally, to isolate only the ribosomal fraction, samples were separated in a sucrose cushion centrifugation and the ribosomal pellets were resuspended in RNC low-salt buffer supplemented with ATP/$BeF_x$. Subsequently, a reference aliquot ($t_0$) was removed from the samples

(complete or only the isolated ribosomal pellets) before initiating degradation of RNCs (0.2 µM) with Proteinase K (Millipore, 2.5 ng/µL) at 4 °C. Aliquots of reactions were removed and quenched at different times by 1:1 mixing with 5 mM PMSF in RNC low-salt buffer, then analysed by SDS-PAGE and immunoblotting. Anti-β-galactosidase antibody (abcam, ab221199) was used to detect all β-galactosidase fragments. Anti-β-galactosidase antibody (abcam, ab106567) was used to detect β-galactosidase fragments with intact N-terminus (i.e., with C-terminal truncations).

### Co-sedimentation assays

For co-sedimentation assays, 1 µM RNCs or empty 70S ribosomes (NEB) were incubated (30 min, 30 °C) with (co-)chaperones (DnaK – 5 µM, TF – 5 µM, DnaJ – 2 µM, GroEL 14-mer 5 µM, GroES 7-mer 10–20 µM) in RNC low-salt buffer. For DnaK co-sedimentation assays the buffer was supplemented with 1 mM ATP. For GroEL/ES co-sedimentation assays, RNCs were first incubated with GroEL (WT or SR1) and/or GroES (20 min, 30 °C) followed by, when indicated, addition of 1 mM ADP or ATP optionally with 1 mM $BeSO_4$ and 10 mM NaF (10 min, 30 °C) as described in Figure captions. Pre-formed $EL:ES_2$ complex was prepared by incubating (10 min, 30 °C) GroEL and GroES in RNC low-salt buffer with 1 mM ATP, 1 mM $BeSO_4$ and 10 mM NaF. For α-lactalbumin competition experiments, 5 µM GroEL and 0.5–24 µM α-lactalbumin (Sigma) were added simultaneously to 5 µM $RNC_{1-510mut}$ and incubated at 30 °C for 20 min in in RNC low-salt buffer. Following incubation of ribosomes or RNCs with (co-)chaperones, the samples were loaded onto a 35% sucrose cushion, and pelleted by centrifugation (264,000 g, 2 h). Following two washes with ice-cold RNC low-salt buffer, the pellet was resuspended at 4 °C. Resuspended pellets from co-sedimentations assays were analysed by SDS-PAGE or proteomics analysis and where indicated used as input for other assays. The sucrose cushion and washing buffer were supplemented with ADP, ATP, $BeSO_4$ or NaF where appropriate.

### Immunoblotting

Following SDS-PAGE, proteins were transferred onto a PVDF membrane using a Trans-Blot Turbo Transfer System (BioRad). Membranes were blocked in PBS-Tween with 5% non-fat milk for 1 h at RT and incubated with appropriate primary antibodies (1:1000 dilution in PBS-Tween with 5% non-fat milk) for 1 h at RT. Following three washes (5 min, RT) with PBS-Tween, membranes were incubated with appropriate HRP-conjugated secondary antibodies (1:10,000 dilution in PBS-Tween with 5% non-fat milk) for 1 h at RT and washed. Membranes were developed by enhanced chemiluminescence using SuperSignal West Pico PLUS Chemiluminescent Substrate (ThermoScientific). Antibodies used for each immunoblot are specified in figure legends. Primary antibodies used in this study are mouse anti-GroEL (abcam, ab82592), rabbit anti-GroES (Enzo, ADI-SPA-210-D), rabbit anti-S2 (Antibodies-Online, ABIN2938988), rabbit anti-L7/L12 (abcam, ab225681), rabbit anti-β-galactosidase raised against the full-length protein (abcam, ab221199) and chicken anti-β-galactosidase raised against a 17 amino acid peptide from near the N-terminus (abcam, ab106567). Primary antibodies were detected with HRP-conjugated goat anti-rabbit (abcam, ab205718), anti-mouse (abcam, ab205719), and anti-chicken (abcam, ab97135) secondary antibodies.

### In vitro translation reactions

In vitro translation reactions (IVTs) were performed using the PURExpress in vitro Protein Synthesis Kit (NEB) supplemented, prior to expression, with 2 units/µL RNasin Ribonuclease Inhibitor (Promega), 5% (v/v) FluoroTect Green Lys (Promega), 10 ng/µL template DNA and, where indicated, 1 µM GroEL 14-mer with or without 3 µM GroES heptamer. Template DNA encoded full-length β-galactosidase either with or without two destabilising mutations I141N, G353D[34]. For expression, IVT reactions were pre-mixed on ice and incubated at 37 °C for 2 h

prior to inhibition of protein synthesis with 2 mM chloramphenicol. Subsequently, β-gal activity was measured by diluting reactions 50x in RNC low-salt buffer with 2.1 mM o-nitrophenyl-β-D-galactopyranoside (oNPG), following absorbance at 420 nm at 25 °C and recording the slope of the progress curve. Soluble and insoluble fractions of the reactions were separated by centrifugation (30 min, 21,000 $g$). The amount of full-length β-gal present in complete reactions and the soluble and insoluble fractions was quantified by separating individual proteins using SDS-PAGE and quantifying the fluorescence of the band corresponding to full-length β-gal which incorporated FluoroTect label during synthesis in IVTs. Quantification of fluorescence was performed in Fiji[74]. Enzyme activity values and levels of soluble protein were normalised to the total amount of full-length β-gal produced in corresponding IVTs. Statistical analysis was performed in GraphPad Prism 9 using one-way ANOVA with Dunnett's multiple comparisons.

### Proteomic quantification of protein content in ribosomal pellets

Protein content in resuspended pellets following co-sedimentation assays was determined as previously described[31] with slight modifications. In short, 10 μg of total protein estimated from the ribosome concentration (based on absorbance at 260 nm) was separated in 8 mm on NuPAGE Bis-Tris gels (ThermoScientific, 1.0 mm, 10–12 wells, 12%) followed by Quick Coomassie Stain (Neo Biotech) staining, band excision and destaining in extraction buffer (50% acetonitrile, 100 mM ammonium bicarbonate, 5 mM DTT, 16 h, 4 °C). Samples were subsequently alkylated (40 mM chloroacetamide, 160 mM ammonium bicarbonate, 10 mM TCEP, 20 min, 70 °C), dehydrated in 100% acetonitrile, air-dried, and digested with trypsin (Promega). Tryptic peptides were loaded onto Evotips (Evosep) and eluted using the 30SPD gradient via an Evosep One HPLC[75] with a 15 cm C18 column into a Lumos Tribrid Orbitrap mass spectrometer (ThermoScientific) via a nanospray emitter (2200 V). Acquisition parameters were set to data-dependent mode with precursor ion spectra acquired at 120,000 resolution followed by higher energy collision dissociation. Raw files were processed in MaxQuant[76] and Perseus[77] with Uniprot *E. coli* reference proteome database and a database for common contaminants. Protein and peptide false detection rates using a decoy reverse database were set to 1%. Quantification of proteins was achieved using iBAQ (intensity-based absolute quantification) and values were normalised to the average intensity of 70S ribosomal proteins. Statistical analysis was performed in GraphPad Prism 9. Raw and normalised values are listed in Supplementary Data 2.

### Equilibrium HDX-MS analysis of RNCs

HDX-MS analysis of RNC:chaperonin complexes was conducted as previously described[31] with slight modifications. In short, stocks of freshly purified RNCs and RNC:chaperonin complexes (prepared via cosedimentation assays) were diluted to 5–6 μM in RNC low-salt buffer. Additionally, stocks of full-length native β-galactosidase, empty 70S ribosomes (NEB), GroEL and pre-closed ATP/BeF$_x$-stabilised EL:ES$_2$ were prepared as controls. Deuterium labelling was initiated by 1:10 dilution of the stock solution in deuteration buffer (10 mM HEPES-NaOD, pD 7.5, 30 mM KOAc, 12 mM Mg(OAc)$_2$, 1 mM DTT, RiboLock RNase inhibitor, 97% D$_2$O). Following labelling at 25 °C for 10 or 100 s, the reaction was quenched with an equal volume of ice-cold quench buffer (100 mM sodium phosphate, pH 1.4, 4 M guanidium hydrochloride, 10 mM TCEP) lowering the pH to 2.5. Digestion was performed using agarose-immobilised pepsin (100 s, 10 °C). The sample was then filtered (0.22 μm PVDF filters, 13,000 $g$, 15 s, 0 °C) and snap frozen in liquid nitrogen for short-term storage. The same protocol was followed to prepare undeuterated controls, except the deuteration buffer was replaced by a H-based buffer (10 mM HEPES-NaOH in H$_2$O, pH 7.5, 30 mM KOAc, 12 mM Mg(OAc)$_2$, 1 mM DTT, RiboLock RNase inhibitor). Note that the RNC low-salt buffer and deuteration buffer were supplemented with 1 mM ATP, 1 mM BeSO$_4$ and 10 mM NaF when labelling the GroEL:ES$_2$:RNC complex and relevant control samples.

Frozen samples were thawed and injected into an Acquity UPLC M-class system with the cooling chamber containing the chromatographic columns kept at 0 ± 0.2 °C. Peptides were trapped (4 min, 200 μL/min) on a C4 trap column (Acquity BEH C4 Van-guard precolumn, 2.1 mm × 5 mm, 1.7 μm, Waters) and separated on a reverse phase Acquity UPLC HSS T3 column (1.8 μm, 1 mm × 50 mm, Waters) at a flow rate of 90 μL/min using a 25 min 3–30% gradient of acetonitrile in 0.1% formic acid. Analysis was performed using a Waters Synapt G2Si HDMS$^E$ instrument in ion mobility mode, acquiring in positive ion mode over a range of 50–2000 m/z with the conventional electrospray ionisation source operated at a source temperature of 80 °C with the capillary set to 3 kV.

MS$^E$ data were processed using Protein Lynx Global Server (PLGS, Waters) to identify peptides in the undeuterated control samples using information from a non-specific cleavage of a database containing sequences of *E. coli* β-galactosidase, GroEL, GroES, 70S ribosomal proteins as well as porcine pepsin. PLGS search was performed using energy thresholds of low = 135 counts and elevated = 30 counts. Peptides identified by PLGS were subsequently filtered and processed in DynamX (Waters) with filters of minimum products per amino acid of 0.05 and minimum consecutive products of 1. All spectra were manually inspected, and poor-quality assignments were removed. Relative deuterium uptake in Da was calculated by subtracting the centroid mass of undeuterated peptides from those of deuterated peptides. Fractional uptake was calculated by dividing the relative uptake by the theoretical maximum for each peptide, equal to n-1, where n is the peptide length excluding prolines. Mean values of deuterium uptake are reported as relative as they are not corrected for back-exchange. Uptake differences >0.5 Da at any time point were considered to be meaningful. All uptake data, peptide coverage maps, and a summary of experimental conditions[78] are shown in Supplementary Data 1. Since difference data were highly similar between the two deuterium exposure times (10 and 100 s), we show data for only the 100 s time point in the main figures.

### Crosslinking mass spectrometry

Crosslinking of ribosome:chaperonin or RNC:chaperonin complexes was conducted as described previously[31] with slight modifications. Reaction conditions were previously optimised and carefully controlled to avoid over-crosslinking. Purified GroEL and GroES were first buffer-exchanged into RNC low-salt buffer using Micro Bio-Spin 6 Columns (Biorad) to remove any traces of Tris-HCl. Empty ribosomes (NEB) or RNCs freshly purified from ΔTF *E. coli* were subsequently diluted in RNC low-salt buffer to a final concentration of 1–2 μM and incubated (30 min, 30 °C) with 5 μM GroEL with or without 10 μM GroES. Samples with GroES were additionally incubated with 1 mM ATP, 1 mM BeSO$_4$ and 10 mM NaF (10 min, 30 °C). Complexes were then cooled to 25 °C and 1 mM DSBU (ThermoScientific) was added to initiate the crosslinking reaction (1 h, 25 °C) followed by quenching with 20 mM Tris-HCl, pH 7.5. Finally, the ribosomal fraction was pelleted via sucrose cushion centrifugation (264,000 g, 2 h, 4 °C) and resuspended in RNC low-salt buffer.

Crosslinked samples were subsequently reduced (10 mM DTT), alkylated (50 mM iodoacetamide) and digested with trypsin. Tryptic peptides were fractionated using a high pH reverse phase chromatography (gradient of acetonitrile in 10 mM NH$_4$HCO$_3$, pH 8, TARGA C18 columns, Nest Group Inc.), lyophilised, and resuspended in 1% formic acid and 2% acetonitrile. Samples were then subjected to nano-scale capillary LC-MS/MS using the Vanquish Neo UPLC (ThermoScientific Dionex), a C18 PepMap Neo nanoViper trapping column (5 μm, 300 μm × 5 mm, ThermoScientific Dionex), and an EASY-Spray column (50 cm × 75 μm ID, PepMap C18, 2 μm particles, 100 Å pore size, ThermoScientific). Peptides were eluted with a gradient of acetonitrile

over 90 min. Analysis was performed using a quadrupole Orbitrap mass spectrometer (Orbitrap Exploris 480, ThermoScientific) with a nano-flow electrospray ionisation source. Acquisition parameters were set to data-dependent mode using a top 10 method, recording a high-resolution full scan ($R = 60,000$, m/z 380-1800) followed by higher energy collision dissociation (stepped collision energy 30 and 32% Normalised Collision Energy) of the 10 most intense MS peaks, excluding ions with precursor charge state of 1+ and 2 + . The fragment ion spectra were acquired at a resolution of 30,000 and a dynamic exclusion window of 20 s was applied.

MeroX[79] was used to search the raw data. Searches were performed against an ad hoc protein database containing the sequences of GroEL, GroES, β-galactosidase and 70S ribosomal proteins as well as a set of randomised decoy sequences generated by MeroX with the minimum peptide length set to 5 amino acids, maximum number of missed cleavages set to 3 and False Discovery Rate cut-off set to 5%. Variable modifications were set to carbamidomethylation of cysteine (mass shift 57.02146 Da) and methionine oxidation (mass shift 15.99491 Da). DSBU modified fragments correspond to 85.05276 Da and 111.03203 Da (precision: 5 ppm MS and 10 ppm MS/MS). Crosslinks to non-native N-terminal sequences (present in purified RNCs because of N-terminal purification tag cleavage) and crosslinks with scores below 50 were disregarded. xiVIEW online tool[80] was used to visualise crosslinks on linear domain diagrams. Detected crosslinks are listed in Supplementary Data 3.

### Negative stain electron microscopy

RNC:GroEL and EL:ES$_2$:RNC complexes prepared via co-sedimentation assays described above were diluted to 12 nM in RNC low-salt buffer. 4 μL were applied to copper grids with 300 mesh carbon film (EM Resolutions) that had been glow discharged (25 mA, 60 s) in the Glo-Qube Plus Glow Discharge System (Quorum Technologies). After 1–2 min of incubation, excess sample was removed with filter paper and 2% uranyl acetate was applied three times for 20 s. Excess stain was removed with filter paper and grids were air-dried. Around ~500 micrographs were collected for each dataset with a pixel size of 4.3 Å/pixel, and defocus range of −1 to −2.2 μm on a FEI Tecnai Spirit 120 kV TEM. Data-processing was performed using cryoSPARC[81]. Following CTF estimation, particles were automatically picked using the blob picker and classified in 2D. 2D classes corresponding to GroEL particles were used as templates for a second auto picking. For each analysed dataset, between 30,000 and 40,000 GroEL or GroEL:ES$_2$ particles were selected after at least two rounds of 2D classification. Particles were subsequently used to generate an initial 3D model. After at least two rounds of 3D classification, two 3D classes in each dataset were selected for 3D refinement. The number of particles in each class is stated in each figure. EM map docking, visualisation and figures were prepared using UCSF ChimeraX[82].

### Reporting summary

Further information on research design is available in the Nature Portfolio Reporting Summary linked to this article.

## Data availability

HDX-MS data are available in Supplementary Data 1, quantitative proteomics in Supplementary Data 2, and XL-MS data in Supplementary Data 3. The mass spectrometry proteomics data have been deposited to the ProteomeXchange Consortium via the PRIDE partner repository[83] with the following dataset identifiers: Composition of ribosomal pellets: PXD054251. HDX-MS: PXD054376. XL-MS: PXD054379. Source data are provided with this paper.

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

## Acknowledgements

We thank F.U. Hartl (MPI Biochemistry) for the plasmids encoding TF, DnaK, DnaJ and GroEL SR1; John Christodoulou (University College London) for the Δtig E. coli strain; Svend Kjaer for preparing GFP-clamp resin; Grant Pellowe and Stephane Mouilleron (Francis Crick institute) for purified 3 C and TEV proteases; Santosh Shivakumaraswamy for purified full-length β-gal; Tomas Voisin, Milos Cvetkovic and Qu Chen for assistance with the nsEM experiments; Christelle Soudy (Francis Crick Institute Chemical Biology STP), Grant Pellowe, Santosh Shivakumaraswamy and Karim El-bouri for help preparing immobilised pepsin; Steven Howell for proteomic analysis of the ribosomal pellets; and all members of the Protein Biogenesis Lab (Francis Crick Institute) for help and discussion. This work was supported by funding from the UKRI (Folding-Map, EP/X020428/1) to D.B., and the Francis Crick Institute which receives its core funding from Cancer Research UK (CC2025, CC2059), the UK Medical Research Council (CC2025, CC2059), and the Wellcome Trust (CC2025, CC2059) to D.B and R.I.E. R.I.E. acknowledges funding from a Royal Society Wolfson Fellowship and The Crick Chris Banton Translation Fund.

## Author contributions
A.R. performed most of the experiments, analysed the data and wrote the manuscript together with D.B. S.L.M. and J.M.S. collected and processed the XL-MS data. G.J., J.H. and R.I.E. purified GroEL and GroES. A.P. performed pelleting experiments with mutant GroEL. D.B. conceived and supervised the project.

## Funding

## Competing interests
The authors declare no competing interests.
