## [Transparent Peer Review file · Nature Communications]

GroEL/ES chaperonin unfolds then encapsulates a nascent protein on the ribosome

Corresponding Author: Dr David Balchin

Version 0:

Reviewer comments:

Reviewer #1

(Remarks to the Author)

GroEL/ES commonly promotes post-translational protein folding. Thereby it transiently encapsulates its substrate within a central chamber of GroEL with GroES acting a lid of the chamber. Recently it was described, that GroEL/ES also acts co-translationally and recognizes nascent chains which are still bound to the ribosome. In their manuscript, Roeselova et al. thoroughly describe the mechanism of GroEL/ES acting on nascent chains bound to the ribosome. They show that GroEL binds to the nascent chain, partially encapsulating it in its cavity. Additionally, engaging binding sites on the outside of the ring, GroEL directly attaches to the ribosome itself. Interestingly, GroES still is able to act as lid on the substrate bound cavity revealing unexpected plasticity of the GroEL/ES-substrate complex. Competition experiments with other chaperones attaching to nascent chains show that Trigger factor and GroEL/ES can act together on longer nascent chains, while DnaK seems to act antagonistic to GroEL.

Overall, the authors present a comprehensive, very sound and convincing data describing the mechanism of substrate folding of GroEL/ES on the ribosome in detail. The conclusions drawn are valid.

Specific Comments:

- Fig. 4/5/6: The description of the experimental approach and the respective results (especially Fig 4) is hard to follow for a reader from a broader audience. A description of the rationale of using ADP/BeFx or ATP/BeFx seems to be missing (to explain which condition traps what complex).
- Fig 5: According to the data presented, GroEL/ES is able to bind to RNC (ribosome nascent chain complexes) in parallel as both cavities can be filled with substrate (Fig. 5D). Would this indicate that two ribosomes can be attach in parallel to GroEL/ES? In the following mechanistic models, the authors (leave out and) do no comment on a parallel processing of nascent chains in both cavities. Please address/explain in more detail.
- Fig. 4/5/6: As it is highly surprising that GroES acts on the RNC-bound cis cavity, using an additional (maybe orthogonal) control like a GroEL single ring mutant (in some of the experiments) would further strengthen the presented data set.
- Fig. 7/8: While the authors address the mechanism of GroEL/ES on the ribosome in detail, it remains elusive how this mechanism and co-translational folding of GroEL/ES compete. The authors might like to additionally address the competition of a free, destabilized substrate protein and RNC. This might allow to address this open issue and the preferences in the binding affinities of GroEL/ES.
- Beta-galactosidase co-translationally assembles into its active, tetrameric form. The authors might like to discuss whether such specific assembling needs would correlate/explain specific NC assisting chaperone systems.

Reviewer #3

(Remarks to the Author)

This paper by Roeselová et al. uses chemical crosslinking, structural proteomics, electron microscopy and biochemical reconstitution of GroEL-associated ribosomes, expressing full-length and truncated nascent chains (NC) of beta galactosidase. It addresses how the E. coli chaperonin, GroEL, recognizes, and together with GroES, modulates misfolded structures in ribosome-tethered nascent proteins. The authors show that GroEL and Trigger Factor can simultaneously engage long nascent chains, whereas GroEL and DnaK exhibit mutually exclusive binding to the emerging NC.

Roeselová et al., show that upon binding within the upper cavity of GroEL, NCs become structurally “destabilized”, i.e. unfolded (at least partially). Although mirroring earlier findings from a proton exchange study of GroEL and ATP action on a ribosome-free pre-denatured enzyme (Lin and Rye 2004 10.1016/j.molcel.2004.09.003), the new evidence is that middle segments within a NC, which in the narrow channel of the ribosome are presumably still without a structure, readily acquire native or misfolded structures, as soon as they exit the ribosome. GroEL binding, in collaboration with ATP-induced GroES-binding, is shown to structurally destabilize the misfolded structures in the emerging NCs, which, once locally released within the chaperonin cavity, are given a chance to refold into more native structures.

This paper is experimentally well designed. It contributes considerably to a better understanding of the mechanism by which, alongside DnaK and Trigger factor, GroEL-GroES chaperonins engages and encapsulates NCs co-translationally, ultimately leading to their local native refolding. Once the authors have responded to our comments, this paper deserves publication in Nature Communications.

Minor findings and remarks:

1) What is the role of the double GroES-capped football species in NC-folding ?

Interestingly, the data show that ribosome-tethered polypeptides bind the inside of the GroEL cavity and become encapsulated in mostly double-capped GroES7-GroEL14-GroES7 complexes, indicating a role for this particular species, in rescuing co-translational misfolding. Whereas these so-called “football” GroES7-GroEL14-GroES7 complexes have long been considered to be in vitro artifacts (Engel et al., 1995 doi: 10.1126/science.7638600), a recent in situ electron tomography study by Wagner et al., 2024 (doi: 10.1038/s41586-024-07843-w.) has shown that up to 40% of the chaperonin complexes are footballs in intact cells, some of which were observed close to ribosomes, as also shown in Fig 5C. The specific function in the chaperonin cycle of single-capped complexes in binding misfolded NC substrates and releasing refolded NCs, and of the double-capped football complexes in destabilizing misfolded NC structures, should be specifically addressed in the discussion.

2) GroEL can act upon separate domains in the middle of a long polypeptide.

This paper provides new information on how chaperonin, thus far generally thought to act globally upon whole polypeptides that are smaller than ~55 kDa, may also act locally upon longer polypeptides, on local misfolded structures/domains flanked by other domains that may not necessitate chaperone processing. Roeselová et al., show that a misfolded domain can be individually “destabilized” (unfolded) within the GroEL cavity under a sealed GroES7 cap while, for lack of space in the cavity, its flanking sequences must hang outside the doubled capped chaperonin complex. This is implying that extended polypeptide linkers between domains can go through the tight GroEL-GroES interphase, apparently without disturbing the effective capping of the chaperonin cavity. This has been shown in part in prior works (which should be cited here). For example, with a pre-denatured GFP-rhodanese fusion substrate, the rhodanese was found to be unfolded/refolded separately within the GroEL cavity under GroES, whereas for lack of space, the fused GFP domain must have been hanging outside the capped chaperonin cavity (doi: 10.1073/pnas.0700070104).

3) Is GroEL’s cavity preventing aggregation of unfolded substrates or is it unfolding misfolded substrates (or both)?

Roeselová et al., show here that GroELs may effectively act upon misfolded structures/domains positioned amidst a lengthy NC. This seems to be in contradiction with the general belief, since 1989, that by entrapping aggregation-prone unfolded polypeptides in its cavity, GroELs would merely prevent wrong interactions with outside polypeptides. The sequestration and confinement in the GroEL cavity would thereby promote unfolded species, to spontaneously reach their native state. An alternative to the above suggestion by the authors, which should be discussed in the paper, is that upon GroEL binding and ATP-hydrolysis, bound misfolded segments (domains) from an emerging NC substrate, may benefit from being transiently unfolded by GroEL and released to spontaneously refold to the native state, within the chaperonin’s cavity, under the GroES cap. Mechanistically, however, it was not demonstrated here that the benefit from the confinement in the cavity lays in prevention of wrong interactions and aggregations with outside polypeptides.

4) What does proximal TIG binding and distal GroEL binding to Ribosome-tethered RCs means?

Deuerling et al., 1999 have initially shown that, TIG and DnaKJ collaborate and complement each other at the ribosome exit, at promoting the proper folding of emerging NC domains (doi: 10.1038/23301). Yet, evidence has accumulated in recent years, showing that GroELs too, may assist in NC folding (Ying et al., 2005, OI 10.1074/jbc.M500364200) and can in part alleviate the phenotype severity of delta DnaKJ and/or delta TIG mutants. Here, Roeselová et al., show that DnaK binding to NCs competes with GroEL binding and they suggest that GroEL and DnaK are mutually antagonistic (Fig 8). This however, may be a misleading interpretation, given that like GroEL/GroES, DnaK/DnaJ is a chaperone that targets misfolded polypeptides and uses energy from ATP hydrolysis to unfold them. Competing binding and unfolding activities by DnaKJ’s would thus be expected to strongly reduce the observed binding and unfolding activities of GroELs, thereby leading an apparent antagonism, which in fact could result from overlapping redundant unfolding/refolding activities, specifically targeted to misfolded NC substrates.

5) The affinity and specificity of GroEL for binding to ribosomes can be better assessed.

There is no information on the specificity and affinity of the interactions that were found by crosslinking and Mass spectroscopy, between the GroEL’s external surface and the NC-tethered-ribosome. The authors should discuss the possibility that the observed crosslinked segments are not specific and merely result from frequent close random encounters between NC-tethered ribosomes and the dangling NC-bound GroELs complexes.

6) Is GroEL-induced NC unfolding only due to confinement in the cavity or may it also be due to entropic pulling?

The authors states that GroEL “locally unfolds the NC, which then refolds”. Does it unfold only misfolded polypeptides? If GroEL unfolded native protein structures, this would be surprising and counter intuitive. The authors may want to discuss the

possibility that when a tethered nascent chain has just emerged from the ribosome and happened to misfold, the binding to a GroELS complex to a distal misfolded N-terminal segment of that NC, and the resulting dangling of the bulky chaperonin complex for a relatively long time, it expected to apply an unfolding force of entropic origin on the whole emerged NC. Thus, distal GroELS binding could cause the partial enfolding of misfolded segments that are not necessarily within the cavity sealed under GroES, but also tethered outside between the ribosome and the chaperonin. This idea of unfolding also by entropic pulling is strengthened by the observation that Trigger factor (TIG) doesn't compete with GroEL for binding to the NC, likely because TIG was found bound closer to the ribosome exit, to NC segments that had no time and space to misfold, than GroEL to more distal NC segments that had more time and space to misfold.

TIG does not use ATP hydrolysis to assist the native shaping of the emerging NC sequences into native domains. TIG was not observed here to structurally destabilize (unfold) emerging NC sequences. In contrast, GroELS, which can use ATP to unfold misfolded ribosome-free polypeptides (Priya et al., 2013 doi:10.1073/pnas.1219867110) was observed here to structurally destabilize (unfold) emerging NC sequences. How would the authors distinguish between a direct, local binding in the GroEL cavity and ATP-fueled unfolding of an emerging misfolded NC segment, leading to its native refolding under GroES, from an indirect unfolding by entropic pulling by a dangling, distally anchored GroELS complex, leading to the native refolding outside the cavity of an NC segment situated between the ribosome and the GroELS complex? Please discuss this question in the paper.

Figure 1 line 71: This paper highlights the importance of double capped GroES7-GroEL14-GroES7 species. The model of the GroELS cycle in figure 1 should therefore include a fourth football intermediate that would precede the last, single-capped, product-release species (right).

Figure 8 line 367: Typo - Mutant instead of WT

Methods line 450: Stalling sequence from the paper of Cymer et al seems to be WWWPPIRGSP and not WWWPRIRGPP. Did the authors mutated the Pro-6 → Arg and the Ser-2 → Pro to make it more potent? Did the authors test the two sequences?

Proteomics analyses. Given the great variation, up to 3 folds of the total IBAQs in each separate sample, the authors should best normalize in a given sample, each IBAQ from GroEL and GroES with the total IBAQs presents in that sample (rIBAQ = relative IBAQ). Thus, standard errors would be smaller within the triplicates.

This method can be found, for example, in the paper by Krey and colleagues

<https://pmc.ncbi.nlm.nih.gov/articles/PMC3946283/>

As an example, by using rIBAQ for the results of the Fig4 C (Left), appear with smaller SEs (right):

Dataset with the RAW IBAQ:

Sheet line 183 for the Uniprot ID: Large ribosomal subunit assembly factor BipA misses for ID P0DTT0.

It is nice to see that triplicate with depleted GroES show virtually no contaminants of GroES. A small check of the ratio between GroES and GroEL gives around between 1.5-2/1 ratio in the experiments by using rIBAQ and MW of each protein, which is consistent with the results. It is noteworthy, however, that in *E. coli* cells, quantitative mass spectroscopy finds equimolar GroEL and GroES protomers (Fauvet et al., 2021 DOI: 10.3389/fmolb.2021.653073).

Reviewer #4

(Remarks to the Author)

Reviewer #5

(Remarks to the Author)

This paper uses a range of methods to address an interesting and important question, namely the nature and consequences of the interaction of the chaperonin GroEL with nascent chains on the ribosome. This is an important question to address as the consensus view is often that TF and DnaK/J are the key chaperones in this regard, whereas GroEL/ES is often thought (and portrayed in many diagrams of proteostasis mechanisms) as only acting post-translationally, although there is already evidence that this is an incomplete view.

In the paper a range of mostly biophysical and imaging methods are used to show convincingly that GroEL does indeed interact with ribosomes, both with and without NCs present on them. The nature of these interactions are probed using cross-linking and deuterium exchange experiments, and some of the complexes formed are investigated using EM and image analysis. The role of the cochaperonin GroES is also examined, and finally the relationships with the other ribosome-interacting chaperones is studied.

Overall I found the paper to be very readable and I am in agreement with the conclusions and the interpretation of the data obtained. There were several places where questions that I had noted down were dealt with in subsequent sections. The authors are to be congratulated on a well-designed and thorough series of experiments.

I have a few minor comments.

1. There's an on-going disagreement in chaperone papers as to whether the proteins on which they act should be referred to as clients or substrates. I strongly prefer clients (GroEL does have a substrate: it is ATP) because of the non-enzymic nature of chaperone-assisted protein folding, and I'd prefer the authors to change their usage, but I'm also aware that the use of the word substrates has quite broad acceptance, so I can't insist.
2. Line 106: I'm not convinced that the authors have formally shown suppression of aggregation here, and I can't see evidence for it in Figure S1F or S1G. (Incidentally, do these figures show the results of three independent repeats? It's not clear from the legend). Can this be made clearer?
3. Line 112 - I'd welcome some brief discussion of how cross-linking artifacts were avoided or reduced in the use of DSBU. (This could be a single sentence added to the relevant methods section). My (non-expert) understanding was that these can be a problem when using homo-bifunctional NHS-esters.
4. The high protection of the C-terminal tail (lines 147-149) is fascinating in view of the apparent dispensability of this region in vivo. Maybe the authors could comment on this in the Discussion.
5. Line 187 and Figure 3F and its analysis: it would be good to have some indication of the relative numbers of free and associated GroEL particles on the EM grids. On its own I wouldn't have found this data very convincing (GroEL is notoriously sticky, so there's a real danger of artifacts arising during sample preparation), though I don't think this is a problem because of all the other evidence from different methods. Nevertheless, how do the authors rule out chance association on the grid, either simply because the particles fall in the same position, or because of a non-physiological interaction?
6. The order of addition experiments shown in Figure 4A very clearly show the "football" form of the GroEL.ES complex, and this is justified by the image reconstructions shown in Figure 5D. However, the model for in vivo activity shown in Figure 8E shows GroEL.ES as a bullet. There is some discussion in the text about how the NC could bind to the football (ie EL.ES2) complex, which is interesting and relevant, but I was left unclear as to whether the authors think EL binds to the NC as EL alone, subsequently binding groES to the trans ring, or as preformed EL.ES1 complex. Their data does not fully resolve this issue (which would be hard anyway as it's very hard to access the functional complexes in vivo, as opposed to making them under controlled lab conditions) but I think the discussion could be clearer on this point.

Version 1:

Reviewer comments:

Reviewer #1

(Remarks to the Author)

In their revised manuscript, the authors answered all my questions and concerns. The manuscript improved, is acceptable and now indeed presents an impressive and highly interesting contribution to the field.

Reviewer #3

(Remarks to the Author)

I am nearly fully satisfied by the amendments and corrections that the authors have introduced in their manuscript, which followed most of my recommendations.

I have only one comment/request about a systematic modification that was introduced to satisfy a recommendation by reviewer 5, who claimed that polypeptides are not GroEL substrates, and thus "strongly" preferred that the authors will use the term "clients".

In stark opposition, it can be claimed that GroEL is a (class 5) polypeptide unfolding enzyme that can process a large excess of misfolding polypeptide substrates and convert them into an excess of low-affinity, natively refolded polypeptide products. This claim is based on the experimental evidence that in the presence of ATP and equimolar GroES7, each GroEL7 ring acts as the active site of a polypeptide unfoldase nanomachine, whose conformational changes, resulting from ATP-binding, GroES7-binding and ATP-hydrolysis, can iteratively unfold and consequently lead to the native refolding of dozens of misfolding (MDH) substrates, as they form. This is shown under heat-denaturing conditions, where the native MDH reporter enzyme is thermolabile. It is unstable and readily seeks to unfold and misfold again (see Goloubinoff et al., Nature chemical biology 14 (4), 388-395). Thus, the term "client" that was traditionally used for the last 35 years to avoid suggesting that GroEL might be an ATP-fueled enzyme acting on misfolded polypeptide substrates, became obsolete and is now misleading.

Reviewer #4

(Remarks to the Author)

Reviewer #5

(Remarks to the Author)

I've looked at the revised manuscript and am satisfied that the points that I raised have been fully dealt with by the authors, by modifications in the text and some additions to the cited literature.

Reviewer #6

(Remarks to the Author)

The manuscript has already been extensively reviewed, and I share the other reviewer's enthusiasm for the discoveries and the thoroughness with which the experiments were carried out. These findings significantly expand our understanding of the role of GroEL/ES in co-translational folding of nascent chains. The authors made extensive revisions to address all the reviewer feedback, in my opinion, in a satisfying fashion.

I specifically evaluated the mass spectrometry experiments (HDX and xL-MS), which were performed and reported very meticulously. I compliment the authors for their extensive supplementary tables reporting the processed proteomics data.

There are just a few minor comments that the authors can easily address:

- Line 87-89: "GroEL binding to RNC1-510, exposing 2½ domains of β-gal, could be further stabilized by mutating the NC to interfere with native folding of domains 1 and 3 (I141N/G353D)". This sentence is difficult to understand. I suggest revising or breaking it up.
- For non-experts, it would be helpful to mention briefly that the DSBU cross-linker is homobifunctional, targeting lysine residues. What is the distribution of surface lysine residues in the GroEL cavity vs. exterior surface?
- Most HDX-figure panels show data for the 100 s time point for deuterium uptake, even though the authors also collected a shorter 10 s time point. It would be helpful to briefly explain how the authors settled on the 100s time point to measure deuterium uptake.
- For Fig. 7B, the legend describes that both HDX time points are shown: 10 s (in grey) and 100 s (in black). However, the figure only displays grey bars.
- Reference 56 should be updated from the bioRxiv preprint to the final Molecular Cell paper.

Roeselova et al. GroEL/ES chaperonin unfolds then encapsulates a nascent protein on the ribosome

Response to reviewers

We thank the reviewers for their constructive comments and suggestions. A point-by-point response is below, and the revised manuscript is attached with changes highlighted in green.

The major changes are:

- A new experiment demonstrating that single-ring GroEL bound to the RNC can still bind GroES. This supports our interpretation that GroES binds the GroEL cis cavity.
- A new experiment demonstrating that the RNC efficiently competes with a post-translational client for GroEL binding.
- New discussion and revised figures related to the role of symmetric GroEL/ES complexes

In addition to addressing the reviewers' comments, we have edited the manuscript to cite recent work from the Bukau lab identifying nascent chains that bind GroEL (<https://doi.org/10.1038/s41467-025-59067-9>).

Reviewer #1 (Remarks to the Author)

GroEL/ES commonly promotes post-translational protein folding. Thereby it transiently encapsulates its substrate within a central chamber of GroEL with GroES acting a lid of the chamber. Recently it was described, that GroEL/ES also acts co-translationally and recognizes nascent chains which are still bound to the ribosome. In their manuscript, Roeselova et al. thoroughly describe the mechanism of GroEL/ES acting on nascent chains bound to the ribosome. They show that GroEL binds to the nascent chain, partially encapsulating it in its cavity. Additionally, engaging binding sites on the outside of the ring, GroEL directly attaches to the ribosome itself. Interestingly, GroES still is able to act as lid on the substrate bound cavity revealing unexpected plasticity of the GroEL/ES-substrate complex. Competition experiments with other chaperones attaching to nascent chains show that Trigger factor and GroEL/ES can act together on longer nascent chains, while DnaK seems to act antagonistic to GroEL.

Overall, the authors present a comprehensive, very sound and convincing data describing the mechanism of substrate folding of GroEL/ES on the ribosome in detail. The conclusions drawn are valid.

We thank the reviewer for the positive comments and suggestions which have improved the manuscript.

Specific Comments:

- Fig. 4/5/6: The description of the experimental approach and the respective results (especially Fig 4) is hard to follow for a reader from a broader audience. A description of the rationale of using ADP/BeFx or ATP/BeFx seems to be missing (to explain which condition traps what complex).

We have now added additional text contextualising these experiments, and included a description of the rationale for using ADP/BeFx and ATP/BeFx. We now write:

“To address this question, we used nucleotides in combination with metal salts to stabilise specific intermediates in the GroEL/ES reaction cycle (Fig 1A). ADP/BeF_x mimics ATP but is

non-hydrolysable, and stabilises the asymmetric complex with GroEL capped on one end by GroES⁵¹. Adding ATP and BeF_x allows ATP binding and hydrolysis prior to trapping by BeF_x; the resulting complex contains ADP/BeF_x in both rings and is symmetrically capped by GroES⁵¹.”

- Fig 5: According to the data presented, GroEL/ES is able to bind to RNC (ribosome nascent chain complexes) in parallel as both cavities can be filled with substrate (Fig. 5D). Would this indicate that two ribosomes can be attach in parallel to GroEL/ES? In the following mechanistic models, the authors (leave out and) do no comment on a parallel processing of nascent changes in both cavities. Please address/explain in more detail.

This is a good point, raised by all three reviewers. In the revised manuscript we mention the role of symmetric/asymmetric complexes in the introduction and discussion, and amend the schematics in Figure 1 and 8 to include symmetric complexes.

In the introduction we now write:

“GroEL and GroES monomers are expressed at approximately stoichiometric levels⁹, and both asymmetric (one GroES oligomer per GroEL double-ring) and symmetric (two GroES per GroEL) complexes coexist in vivo¹⁰. Thus, one or both cavities can be folding-active.”

In the discussion we now write:

“Our NsEM data revealed substrate density in both rings of EL:ES2 complexes, suggesting that GroEL might bind co- and posttranslational clients simultaneously, or engage two NCs at once. The latter scenario would be facilitated by the clustering of ribosomes in polysomes.”

- Fig. 4/5/6: As it is highly surprising that GroES acts on the RNC-bound cis cavity, using an additional (maybe orthogonal) control like a GroEL single ring mutant (in some of the experiments) would further strengthen the presented data set.

To address this point we have performed GroEL/ES:RNC co-sedimentation experiments using the single-ring mutant of GroEL (GroEL SR1). These data are shown in FigS4C and below.

We find that GroES and GroEL SR1 co-sediment together with the RNC, at similar levels to GroES and wild-type GroEL in the presence of ATP/BeF_x. This is consistent with our conclusion that GroES binds the cis cavity of RNC-bound GroEL.

- Fig. 7/8: While the authors address the mechanism of GroEL/ES on the ribosome in detail, it remains elusive how this mechanism and co-translational folding of GroEL/ES compete. The authors might like to additionally address the competition of a free, destabilized substrate protein and RNC. This might allow to address this open issue and the preferences in the binding affinities of GroEL/ES.

To address this point we have performed GroEL:RNC co-sedimentation experiments in the presence of increasing concentrations of reduced α -lactalbumin, a model substrate that is unfolded and soluble. These data are shown in FigS1D and below.

At a concentration of 5 μM GroEL and 5 μM RNC, we find that > 10 μM α -lactalbumin is required to noticeably reduce GroEL binding to the RNC. Therefore, the RNC is not readily outcompeted by a destabilised post-translational substrate. At intermediate concentrations of lactalbumin, the NC and competing substrate may each occupy one ring of GroEL, a possibility we mention in the revised manuscript.

- Beta-galactosidase co-translationally assembles into its active, tetrameric form. The authors might like to discuss whether such specific assembling needs would correlate/explain specific NC assisting chaperone systems.

In the discussion we now write:

“ β -gal assembles cotranslationally⁵⁹, but assembly initiates only upon emergence of the first 4 domains from the ribosome⁶⁰. For oligomeric proteins such as β -gal, GroEL/ES may play a role in shielding N-terminal domains exposing unsatisfied assembly interfaces, prior to the onset of cotranslational assembly.”

Reviewer #3 (Remarks to the Author)

This paper by Roeselová et al. uses chemical crosslinking, structural proteomics, electron microscopy and biochemical reconstitution of GroELS-associated ribosomes, expressing full-length and truncated nascent chains (NC) of beta galactosidase. It addresses how the E. coli chaperonin, GroEL, recognizes, and together with GroES, modulates misfolded structures in ribosome-tethered nascent proteins. The authors show that GroEL and Trigger Factor can simultaneously engage long nascent chains, whereas GroEL and DnaK exhibit mutually exclusive binding to the emerging NC.

Roeselová et al., show that upon binding within the upper cavity of GroEL, NCs become structurally “destabilized”, i.e. unfolded (at least partially). Although mirroring earlier findings from a proton exchange study of GroEL and ATP action on a ribosome-free pre-denatured enzyme (Lin and Rye 2004 10.1016/j.molcel.2004.09.003), the new evidence is that middle segments within a NC, which in the narrow channel of the ribosome are presumably still without a structure, readily acquire native or misfolded structures, as soon as they exit the

ribosome. GroEL binding, in collaboration with ATP-induced GroES-binding, is shown to structurally destabilize the misfolded structures in the emerging NCs, which, once locally released within the chaperonin cavity, are given a chance to refold into more native structures.

This paper is experimentally well designed. It contributes considerably to a better understanding of the mechanism by which, alongside DnaK and Trigger factor, GroEL-GroES chaperonins engages and encapsulates NCs co-translationally, ultimately leading to their local native refolding. Once the authors have responded to our comments, this paper deserves publication in Nature Communications.

We thank the reviewer for their positive assessment of the manuscript and thoughtful suggestions. In the revised manuscript we include a more detailed discussion of functional states in the GroEL cycle, and the mechanistic origin of NC unfolding by GroEL.

Minor findings and remarks:

1) What is the role of the double GroES-capped football species in NC-folding ?

Interestingly, the data show that ribosome-tethered polypeptides bind the inside of the GroEL cavity and become encapsulated in mostly double-capped GroES7-GroEL14-GroES7 complexes, indicating a role for this particular species, in rescuing co-translational misfolding. Whereas these so-called “football” GroES7-GroEL14-GroES7 complexes have long been considered to be in vitro artifacts (Engel et al., 1995 doi: 10.1126/science.7638600), a recent in situ electron tomography study by Wagner et al., 2024 (doi: 10.1038/s41586-024-07843-w.) has shown that up to 40% of the chaperonin complexes are footballs in intact cells, some of which were observed close to ribosomes, as also shown in Fig 5C. The specific function in the chaperonin cycle of single-capped complexes in binding misfolded NC substrates and releasing refolded NCs, and of the double-capped football complexes in destabilizing misfolded NC structures, should be specifically addressed in the discussion.

This is a good point, raised by all three reviewers. In the revised manuscript we mention the role of symmetric/asymmetric complexes in the introduction and discussion, and amend the schematics in Figure 1 and 8 to include symmetric complexes.

Note that we generated homogeneous complexes by artificially stabilising the double-capped species using ATP/BeF_x. We therefore cannot draw any conclusions from the absence of single-capped complexes.

In the introduction we now write:

“GroEL and GroES monomers are expressed at approximately stoichiometric levels⁹, and both asymmetric (one GroES oligomer per GroEL double-ring) and symmetric (two GroES per GroEL) complexes coexist in vivo¹⁰. Thus, one or both cavities can be folding-active.”

In the discussion we now write:

“Our NsEM data revealed substrate density in both rings of EL:ES2 complexes, suggesting that GroEL might bind co- and posttranslational clients simultaneously, or engage two NCs at once. The latter scenario would be facilitated by the clustering of ribosomes in polysomes.”

2) GroEL can act upon separate domains in the middle of a long polypeptide.

This paper provides new information on how chaperonin, thus far generally thought to act globally upon whole polypeptides that are smaller than ~55 kDa, may also act locally upon longer polypeptides, on local misfolded structures/domains flanked by other domains that may not necessitate chaperone processing. Roeselová et al., show that a misfolded domain can be individually “destabilized” (unfolded) within the GroEL cavity under a sealed GroES7 cap while, for lack of space in the cavity, its flanking sequences must hang outside the doubled capped chaperonin complex. This is implying that extended polypeptide linkers between domains can go through the tight GroEL-GroES interphase, apparently without disturbing the effective capping of the chaperonin cavity. This has been shown in part in a prior works (which should be cited here). For example, with a pre-denatured GFP-rhodanese fusion substrate, the rhodanese was found to be unfolded/refolded separately within the GroEL cavity under GroES, whereas for lack of space, the fused GFP domain must have been hanging outside the capped chaperonin cavity (doi: 10.1073/pnas.0700070104).

We agree, and now cite Kipnis et al. in the discussion (reference 68).

3) Is GroEL’s cavity preventing aggregation of unfolded substrates or is it unfolding misfolded substrates (or both)?

Roeselová et al., show here that GroELS may effectively act upon misfolded structures/domains positioned amidst a lengthy NC. This seems to be in contradiction with the general belief, since 1989, that by entrapping aggregation-prone unfolded polypeptides in its cavity, GroELS would merely prevent wrong interactions with outside polypeptides. The sequestration and confinement in the GroEL cavity would thereby promote unfolded species, to spontaneously reach their native state.

An alternative to the above suggestion by the authors, which should be discussed in the paper, is that upon GroEL binding and ATP-hydrolysis, bound misfolded segments (domains) from an emerging NC substrate, may benefit from being transiently unfolded by GroEL and released to spontaneously refold to the native state, within the chaperonin’s cavity, under the GroES cap. Mechanistically, however, it was not demonstrated here that the benefit from the confinement in the cavity lays in prevention of wrong interactions and aggregations with outside polypeptides.

We agree that our data support a model whereby transient unfolding is followed by refolding inside GroEL/ES. Although we do not demonstrate that the nascent polypeptide is misfolded prior to GroEL binding, we also agree that it is reasonable to speculate that misfolded states would be resolved via this mechanism. In the discussion we write:

“...local unfolding may resolve misfolded states prior to refolding in the GroEL/ES cage.”

4) What does proximal TIG binding and distal GroEL binding to Ribosome-tethered RCs means?

Deuerling et al., 1999 have initially shown that, TIG and DnaKJ collaborate and complement each other at the ribosome exit, at promoting the proper folding of emerging NC domains (doi: 10.1038/23301). Yet, evidence has accumulated in recent years, showing that GroELS too, may assist in NC folding (Ying et al., 2005, OI 10.1074/jbc.M500364200) and can in part alleviate the phenotype severity of delta DnaKJ and/or delta TIG mutants. Here, Roeselová et al., show that DnaK binding to NCs competes with GroEL binding and they suggest that GroEL and DnaK are mutually antagonistic (Fig 8).

This however, may be a misleading interpretation, given that like GroEL/GroES, DnaK/DnaJ is a chaperone that targets misfolded polypeptides and uses energy from ATP hydrolysis to

unfold them. Competing binding and unfolding activities by DnaKJ's would thus be expected to strongly reduce the observed binding and unfolding activities of GroELs, thereby leading an apparent antagonism, which in fact could result from overlapping redundant unfolding/refolding activities, specifically targeted to misfolded NC substrates.

We agree. In the discussion we now write:

“GroEL and DnaK/J may directly compete for similar binding sites on the NC, or the action of one chaperone system could change the conformation of the NC so that it is no longer recognised by the other.”

5) The affinity and specificity of GroEL for binding to ribosomes can be better assessed.

There is no information on the specificity and affinity of the interactions that were found by crosslinking and Mass spectroscopy, between the GroEL's external surface and the NC-tethered-ribosome. The authors should discuss the possibility that the observed crosslinked segments are not specific and merely result from frequent close random encounters between NC-tethered ribosomes and the dangling NC-bound GroELs complexes.

GroEL cosediments with empty ribosomes lacking a NC (Fig 1D and S1C), and also crosslinks to empty ribosomes (Fig 2D) dependent on the presence of the L7/L12 stalk (Fig S2C,D). This argues that the GroEL:ribosome interaction is not simply a byproduct of tethering via the NC. In addition, our previous XLMS analyses of RNC complexes with Trigger factor, DnaJ and DnaK did not yield crosslinks between chaperones and L7/L12 (10.1016/j.molcel.2024.06.002).

Specific residues on GroEL crosslinked reproducibly to the same sites on L7/L12, when mixed with empty ribosomes or 3 different RNCs (Fig 2B and S2A,B), suggesting that the interaction might be specific. The crosslink positions also overlapped with HDX protection on the outer surface of GroEL. However, we certainly do not want to overstate this point or claim that this is a unique binding interface. In the discussion we write:

“We show that GroEL weakly interacts with ribosomes independent of the NC. This occurs via the outer surface of the chaperonin cavity, via sites distinct from those that bind substrates. Although XL-MS suggests that GroEL binds directly to the ribosomal L7/L12 stalk, we do not exclude that a different interface is involved. For example, GroEL binding to ribosomal RNA would not be detected by DSBU crosslinking.”

Only a small amount of GroEL cosediments with ribosomes in the absence of NC, and throughout the manuscript we emphasize that this interaction is weak. We have tried to measure a binding constant using SPR (immobilized GroEL, flow of ribosomes), but the measurements are complicated by the size of the complex which reduces sensitivity away from the sensor surface, and the requirement for high concentrations of ribosomes/RNCs. We are currently working to develop quantitative assays for chaperone binding to RNCs, which will form the basis for a future manuscript.

6) Is GroEL-induced NC unfolding only due to confinement in the cavity or may it also be due to entropic pulling?

The authors states that GroEL “locally unfolds the NC, which then refolds”. Does it unfold only misfolded polypeptides? If GroEL unfolded native protein structures, this would be surprising and counter intuitive. The authors may want to discuss the possibility that when a tethered nascent chain has just emerged from the ribosome and happened to misfold, the binding to a GroELs complex to a distal misfolded N-terminal segment of that NC, and the resulting dangling of the bulky chaperonin complex for a relatively long time, it expected to

apply an unfolding force of entropic origin on the whole emerged NC. Thus, distal GroELS binding could cause the partial enfolding of misfolded segments that are not necessarily within the cavity sealed under GroES, but also tethered outside between the ribosome and the chaperonin. This idea of unfolding also by entropic pulling is strengthened by the observation that Trigger factor (TIG) doesn't compete with GroEL for binding to the NC, likely because TIG was found bound closer to the ribosome exit, to NC segments that had no time and space to misfold, than GroEL to more distal NC segments that had more time and space to misfold.

TIG does not use ATP hydrolysis to assist the native shaping of the emerging NC sequences into native domains. TIG was not observed here to structurally destabilize (unfold) emerging NC sequences. In contrast, GroELS, which can use ATP to unfold misfolded ribosome-free polypeptides (Priya et al., 2013 doi:10.1073/pnas.1219867110) was observed here to structurally destabilize (unfold) emerging NC sequences. How would the authors distinguish between a direct, local binding in the GroEL cavity and ATP-fueled unfolding of an emerging misfolded NC segment, leading to its native refolding under GroES, from an indirect unfolding by entropic pulling by a dangling, distally anchored GroELS complex, leading to the native refolding outside the cavity of an NC segment situated between the ribosome and the GroELS complex? Please discuss this question in the paper.

We do not think that entropic pulling is the key driver of unfolding in the case of β -gal, for two reasons. 1. GroEL unfolds an N-terminal segment of the NC, not the C-terminal part that is between GroEL and the ribosome; the C-terminal part of the NC is already unfolded relative to native Bgal (Fig 7B, peptide 472-479), even in the absence of GroEL. 2. β -gal NC is not unfolded by GroEL:ES₂, which in an entropic pulling mechanism would presumably have a similar effect to GroEL alone.

However, we think the idea that GroEL might unfold other NCs by entropic pulling is compelling, and we now mention this possibility in the discussion section. We now write:

“Alternatively, binding of the bulky GroEL molecule may cause partial unfolding of distal segments of the NC via entropic pulling⁵⁸.”

Figure 1 line 71: This paper highlights the importance of double capped GroES7-GroEL14-GroES7 species. The model of the GroELS cycle in figure 1 should therefore include a fourth football intermediate that would precede the last, single-capped, product-release species (right).

We have changed Fig 1A to include the double-capped species.

Figure 8 line 367: Typo - Mutant instead of WT

Fixed, thank you.

Methods line 450: Stalling sequence from the paper of Cymer et al seems to be WWWPPIRGSP and not WWWPRIRGPP. Did the authors mutated the Pro-6 → Arg and the Ser-2 → Pro to make it more potent? Did the authors test the two sequences?

Based on their mutagenesis screen, Cymer et al. hypothesised that Pro-6 → Arg and Ser-2 → Pro mutations would further improve stalling efficiency of WWWPPIRGSP. We therefore included these two mutations in our stalling sequence. In their paper, Cymer et al. write:

“By simultaneously mutating the three positions -10 to -8 to tryptophan, we were able to generate an AP (WWWPPIRGSP) that is able to stall translation even in the presence of the pulling force elicited by 10 aspartic acid residues (Fig. 3). This AP can, presumably, be made

even stronger by introducing the Ser₋₂ → Pro and Pro₋₆ → Arg mutations, but, because the pulling force from a [10D] stretch is the strongest we have been able to generate so far (16), we cannot determine whether this is the case.”

We have not formally compared WWWPPIRGSP to WWWPRIRGPP, but the latter sequence certainly works well in our hands.

Proteomics analyses. Given the great variation, up to 3 folds of the total IBAQs in each separate sample, the authors should best normalize in a given sample, each IBAQ from GroEL and GroES with the total IBAQs presents in that sample (rIBAQ = relative IBAQ). Thus, standard errors would be smaller within the triplicates.

This method can be found, for example, in the paper by Krey and colleagues <https://pmc.ncbi.nlm.nih.gov/articles/PMC3946283/>

As an example, by using rIBAQ for the results of the Fig4 C (Left), appear with smaller SEs (right):

We appreciate the suggestion and carefully considered this approach. To prepare our plots we already normalised the raw iBAQ values to the average iBAQ for ribosomal proteins in each sample, thus accounting for sample-to-sample variability. This has the advantage of expressing chaperone amount relative to ribosomes, which is particularly meaningful for our study. Although it would be possible to normalise twice (once for all proteins and once for ribosomal proteins), we would prefer not to do so. The majority of each sample consists of ribosomal proteins, and by using only these proteins for normalisation we can be sure that no contaminant proteins contribute to calculation (as warned against by Krey et al).

Dataset with the RAW IBAQ:

Sheet line 183 for the Uniprot ID: Large ribosomal subunit assembly factor BipA misses for ID P0DTT0.

Fixed, thank you.

It is nice to see that triplicate with depleted GroES show virtually no contaminants of GroES. A small check of the ratio between GroES and GroEL gives around between 1.5-2/1 ratio in the experiments by using rIBAQ and MW of each protein, which is consistent with the results. It is noteworthy, however, that in *E. coli* cells, quantitative mass spectroscopy finds equimolar GroEL and GroES protomers (Fauvet et al., 2021 DOI: 10.3389/fmolb.2021.653073).

We now mention this in the introduction when discussing symmetric/asymmetric complexes:

“GroEL and GroES monomers are expressed at approximately stoichiometric levels⁹, and both asymmetric (one GroES oligomer per GroEL double-ring) and symmetric (two GroES per GroEL) complexes coexist in vivo¹⁰.”

Reviewer #4 (Remarks to the Author)

We thank the reviewer for their time and constructive comments.

Reviewer #5 (Remarks to the Author)

This paper uses a range of methods to address an interesting and important question, namely the nature and consequences of the interaction of the chaperonin GroEL with nascent chains on the ribosome. This is an important question to address as the consensus view is often that TF and DnaK/J are the key chaperones in this regard, whereas GroEL/ES is often thought (and portrayed in many diagrams of proteostasis mechanisms) as only acting post-translationally, although there is already evidence that this is an incomplete view.

In the paper a range of mostly biophysical and imaging methods are used to show convincingly that GroEL does indeed interact with ribosomes, both with and without NCs present on them. The nature of these interactions are probed using cross-linking and deuterium exchange experiments, and some of the complexes formed are investigated using EM and image analysis. The role of the cochaperonin GroES is also examined, and finally the relationships with the other ribosome-interacting chaperones is studied.

Overall I found the paper to be very readable and I am in agreement with the conclusions and the interpretation of the data obtained. There were several places where questions that I had noted down were dealt with in subsequent sections. The authors are to be congratulated on a well-designed and thorough series of experiments.

We thank the reviewer for their kind words and positive assessment of our manuscript.

I have a few minor comments.

1. There's an on-going disagreement in chaperone papers as to whether the proteins on which they act should be referred to as clients or substrates. I strongly prefer clients (GroEL does have a substrate: it is ATP) because of the non-enzymic nature of chaperone-assisted protein folding, and I'd prefer the authors to change their usage, but I'm also aware that the use of the word substrates has quite broad acceptance, so I can't insist.

In the revised manuscript we have changed "substrate" to "client", throughout.

2. Line 106: I'm not convinced that the authors have formally shown suppression of aggregation here, and I can't see evidence for it in Figure S1F or S1G. (Incidentally, do these figures show the results of three independent repeats? It's not clear from the legend). Can this be made clearer?

Fig S1F shows the level of soluble protein in IVTs expressing wild-type β -gal in the presence of GroEL or GroEL/ES. Fig S1G shows the same for IVTs expressing the destabilised mutant. The level of soluble β -gal in each reaction is quantified relative to the total β -gal produced (Fig S1E).

WT β -gal is not affected by GroEL/ES (Fig S1E,F), whereas the level of soluble mutant β -gal is increased when the chaperones are present (Fig S1E,G). We opted not to quantify the insoluble fraction because the pellets are very small and we find that the loading is less reproducible. To more accurately reflect what we measure in these experiments, in the

revised manuscript we no longer claim that GroEL suppresses aggregation of mutant β -gal. We now write:

“To understand whether GroEL/ES affects β -gal maturation, we expressed β -gal in a fully-reconstituted in vitro transcription/translation (IVT) system. Supplementing IVTs with GroEL or GroEL/ES did not substantially affect the folding of WT β -gal, but improved the solubility of the I141N/G353D β -gal mutant ~4-fold (Fig S1D-G). In summary, GroEL interacts with ribosome-associated NCs, is sensitive to the conformation of the NC, and increases the soluble yield of destabilised nascent β -gal.

We agree that the contrast of Fig S1G makes it difficult to appreciate the difference in soluble protein when comparing “PURE” to EL or EL/ES. In the revised figure panel, we have adjusted the brightness (of the entire image):

Fig S1F and S1G show the results from three independent IVT reactions. This is now labelled on the figures and described in the legend.

3. Line 112 - I'd welcome some brief discussion of how cross-linking artifacts were avoided or reduced in the use of DSBU. (This could be a single sentence added to the relevant methods section). My (non-expert) understanding was that these can be a problem when using homo-bifunctional NHS-esters.

We assume that the reviewer refers to the possibility of “over-crosslinking” that can occur at high protein concentrations/prolonged reaction times. We tend to have the opposite problem, since RNC concentration is limiting, and we do not detect a large number of crosslinks considering the size of the complex. In general, crosslinking can be controlled by controlling the reaction conditions, which we optimised in our previous study (10.1016/j.molcel.2024.06.002) for similar RNC:chaperone complexes. We also check intra-ribosome crosslinks for consistency with the structure of the ribosome.

In the Methods we now write:

“Crosslinking of ribosome:chaperonin or RNC:chaperonin complexes was conducted as described previously⁴⁶ with slight modifications. Reaction conditions were previously optimised and carefully controlled to avoid over-crosslinking.”

4. The high protection of the C-terminal tail (lines 147-149) is fascinating in view of the apparent dispensability of this region in vivo. Maybe the authors could comment on this in the Discussion.

We now write:

“Moreover, we provide evidence that the disordered C-terminal tails of GroEL directly contact the NC. Although dispensable *in vivo*⁴⁴, the C-tails have previously been shown to contribute to client capture and folding^{42,45–48}, and recent work suggests that they are required for optimal chaperonin function⁴⁹.”

5. Line 187 and Figure 3F and its analysis: it would be good to have some indication of the relative numbers of free and associated GroEL particles on the EM grids. On its own I wouldn't have found this data very convincing (GroEL is notoriously sticky, so there's a real danger of artifacts arising during sample preparation), though I don't think this is a problem because of all the other evidence from different methods. Nevertheless, how do the authors rule out chance association on the grid, either simply because the particles fall in the same position, or because of a non-physiological interaction?

This is a fair point. On reflection we do not think that we can quantify free versus ribosome-associated GroEL on the grids, since we do not know how to estimate a meaningful distance cutoff. Because the NC is dynamic/partially folded, RNC-bound GroEL could in principle be positioned far from the ribosome surface. As alluded to by the reviewer, the complex between GroEL and RNCs is demonstrated by other methods, and we do not think that EM is necessarily best suited to support this claim.

In the revised manuscript we have softened the language related to the position of GroEL near ribosomes on the grid. Related to Fig 3, we now write:

“Negative stain electron microscopy (nsEM) of GroEL:RNC complexes showed particles identifiable as GroEL or ribosomes, and instances of GroEL positioned close to ribosomes”.

Related to Fig 5, we no longer mention the proximity of GroEL/ES to ribosomes, and simply state:

“The position of the nascent chain inside the GroEL/ES cavity was further supported by nsEM of the EL:ES₂:RNC complex. Approximately 90% of GroEL/ES complexes were double-capped EL:ES₂, and analysis of these species resulted in two 3D reconstructions which revealed additional density inside one or both EL/ES cavities.”

6. The order of addition experiments shown in Figure 4A very clearly show the "football" form of the GroEL.ES complex, and this is justified by the image reconstructions shown in Figure 5D. However, the model for *in vivo* activity shown in Figure 8E shows GroEL.ES as a bullet. There is some discussion in the text about how the NC could bind to the football (ie EL.ES₂) complex, which is interesting and relevant, but I was left unclear as to whether the authors think EL binds to the NC as EL alone, subsequently binding GroES to the trans ring, or as preformed EL.ES₁ complex. Their data does not fully resolve this issue (which would be hard anyway as it's very hard to access the functional complexes *in vivo*, as opposed to making them under controlled lab conditions) but I think the discussion could be clearer on this point.

This is a good point, raised by all three reviewers. In the revised manuscript we mention the role of symmetric/asymmetric complexes in the introduction and discussion, and amend the schematics in Figure 1 and 8 to include symmetric complexes.

In the introduction we now write:

“GroEL and GroES monomers are expressed at approximately stoichiometric levels⁹, and both asymmetric (one GroES oligomer per GroEL double-ring) and symmetric (two GroES per GroEL) complexes coexist *in vivo*¹⁰. Thus, one or both cavities can be folding-active.”

In the discussion we now write:

“Our NsEM data revealed substrate density in both rings of EL:ES2 complexes, suggesting that GroEL might bind co- and posttranslational clients simultaneously, or engage two NCs at once. The latter scenario would be facilitated by the clustering of ribosomes in polysomes.”

Regarding the question of whether EL binds the NC as EL alone or EL:ES₁, we see no reason why both would not be possible. In the legend to Figure 8E (the model) we now mention that either complex might initially bind the NC.

Reviewer #1 (Remarks to the Author):

In their revised manuscript, the authors answered all my questions and concerns. The manuscript improved, is acceptable and now indeed presents an impressive and highly interesting contribution to the field.

We thank the reviewer for their positive comments, and for taking the time to help us improve the manuscript.

Reviewer #3 (Remarks to the Author):

I am nearly fully satisfied by the amendments and corrections that the authors have introduced in their manuscript, which followed most of my recommendations.

We thank the reviewer for their time and constructive feedback.

I have only one comment/request about a systematic modification that was introduced to satisfy a recommendation by reviewer 5, who claimed that polypeptides are not GroELS substrates, and thus "strongly" preferred that the authors will use the term "clients". In stark opposition, it can be claimed that GroELS is a (class 5) polypeptide unfolding enzyme that can process a large excess of misfolding polypeptide substrates and convert them into an excess of low-affinity, natively refolded polypeptide products. This claim is based on the experimental evidence that in the presence of ATP and equimolar GroES7, each GroEL7 ring acts as the active site of a polypeptide unfoldase nanomachine, whose conformational changes, resulting from ATP-binding, GroES7-binding and ATP-hydrolysis, can iteratively unfold and consequently lead to the native refolding of dozens of misfolding (MDH) substrates, as they form. This is shown under heat-denaturing conditions, where the native MDH reporter enzyme is thermolabile. It is unstable and readily seeks to unfold and misfold again (see Goloubinoff et al., Nature chemical biology 14 (4), 388-395). Thus, the term "client" that was traditionally used for the last 35 years to avoid suggesting that GroELS might be an ATP-fueled enzyme acting on misfolded polypeptide substrates, became obsolete and is now misleading.

We appreciate the points made by both reviewers. Our view is that neither the term "substrate" nor "client" implies a specific chaperone mechanism.

Reviewer #4 (Remarks to the Author):

We thank the reviewer for their time and constructive feedback.

Reviewer #5 (Remarks to the Author):

I've looked at the revised manuscript and am satisfied that the points that I raised have been fully dealt with by the authors, by modifications in the text and some additions to the cited literature.

We thank the reviewer for their time and constructive feedback.

Reviewer #6 (Remarks to the Author):

The manuscript has already been extensively reviewed, and I share the other reviewer's enthusiasm for the discoveries and the thoroughness with which the experiments were carried out. These findings significantly expand our understanding of the role of GroEL/ES in co-translational folding of nascent chains. The authors made extensive revisions to address all the reviewer feedback, in my opinion, in a satisfying fashion.

I specifically evaluated the mass spectrometry experiments (HDX and xL-MS), which were performed and reported very meticulously. I compliment the authors for their extensive supplementary tables reporting the processed proteomics data.

We thank the reviewer for their positive feedback and constructive comments, as well as for taking the time to step in as a technical expert.

There are just a few minor comments that the authors can easily address:

- Line 87-89: "GroEL binding to RNC₁₋₅₁₀, exposing 2½ domains of β-gal, could be further stabilized by mutating the NC to interfere with native folding of domains 1 and 3 (I141N/G353D)". This sentence is difficult to understand. I suggest revising or breaking it up.

We now write:

"We found that GroEL bound to RNC₁₋₅₁₀ which exposes 2½ domains of β-gal. Binding was further stabilised by introducing NC mutations that disrupt the fold of domains 1 and 3 (I141N/G353D³⁴, Fig 1c,d)."

- For non-experts, it would be helpful to mention briefly that the DSBU cross-linker is homobifunctional, targeting lysine residues. What is the distribution of surface lysine residues in the GroEL cavity vs. exterior surface?

We now write:

"To define the topology of these assemblies, we crosslinked complexes between GroEL and 3 different RNCs using disuccinimidyl dibutyric urea (DSBU) and identified crosslink sites using mass spectrometry (Fig 2a). DSBU is a homobifunctional crosslinker that preferentially targets lysine residues."

Surface lysine residues are approximately evenly distributed between the cavity (13 Lys per monomer) and exterior (14 Lys per monomer) of GroEL. We now state this in the manuscript. The positions of cavity/exterior lysine residues that crosslink to RNCs are shown in Figure 2.

- Most HDX-figure panels show data for the 100 s time point for deuterium uptake, even though the authors also collected a shorter 10 s time point. It would be helpful to briefly explain how the authors settled on the 100s time point to measure deuterium uptake.

Uptake differences were very similar when calculated using either time point. We therefore chose to show just one time point to improve the clarity of the figures. Data from both time points are shown in the supplementary table. In the Methods section we now write:

"Since difference data were highly similar between the two deuterium exposure times (10 and 100 s), we show data for only the 100 s time point in the main figures."

- For Fig. 7B, the legend describes that both HDX time points are shown: 10 s (in grey) and 100 s (in black). However, the figure only displays grey bars.

Fixed, thank you.

- Reference 56 should be updated from the bioRxiv preprint to the final Molecular Cell paper.

Well spotted, thank you. Ref. 56 was a duplicate of ref 31 and has now been removed.

As an example, by using rIBAQ for the results of the Fig4 C (Left), appear with smaller SEs (right):

Dataset with the RAW IBAQ:
Sheet line 183 for the Uniprot ID: **Large ribosomal subunit assembly factor BipA** misses for ID P0DTT0.